# Surfactant-free interfacial growth of graphdiyne hollow microspheres and the mechanistic origin of their SERS activity

Lu Zhang[1], Wencai Yi[2], Junfang Li[1], Guoying Wei[3], Guangcheng Xi[1]✉ & Lanqun Mao[4]✉

As a two-dimensional carbon allotrope, graphdiyne possesses a direct band gap, excellent charge carrier mobility, and uniformly distributed pores. Here, a surfactant-free growth method is developed to efficiently synthesize graphdiyne hollow microspheres at liquid–liquid interfaces with a self-supporting structure, which avoids the influence of surfactants on product properties. We demonstrate that pristine graphdiyne hollow microspheres, without any additional functionalization, show a strong surface-enhanced Raman scattering effect with an enhancement factor of $3.7 \times 10^7$ and a detection limit of $1 \times 10^{-12}$ M for rhodamine 6 G, which is approximately 1000 times that of graphene. Experimental measurements and first-principles density functional theory simulations confirm the hypothesis that the surface-enhanced Raman scattering activity can be attributed to an efficiency interfacial charge transfer within the graphdiyne-molecule system.

As an analysis technique, surface-enhanced Raman scattering (SERS) spectroscopy can provide abundant information about the fine structure and chemical composition of materials down to the single-molecule level[1,2]. SERS has a wide range of applications in the fields of structure characterization, trace analysis, disease diagnosis, and biosensing[3–5], of which SERS substrates are critical because the Raman signals of a molecule are usually weak and need to be significantly enhanced by two well-known mechanisms, namely, the electromagnetic mechanism (EM) and chemical mechanism (CM)[6,7]. Traditional SERS substrates can be divided into two broad categories: noble-metal substrates and semiconductor substrates. Noble-metal substrates, such as Au and Ag, can take advantage of localized field-induced surface plasmon resonance (SPR), especially the emergence of a large number of "hot spots" (e.g., high-intensity electromagnetic field regions formed at nanoscale gaps), resulting in dramatic enhancement of SERS signals[8–10]. Cheaper metals such as copper and aluminium with suitable nanostructures have also been found to have SERS activity[11–16]. Semiconducting SERS substrates, such as $WO_3$, ZnO,

and $MoO_2$, have been shown to exhibit CM enhancement, in which interfacial charge transfer (ICT) plays a dominant role[17–19]. Carbon materials have also been demonstrated to possess SERS sensing properties and have attracted increasing attention due to their abundant surface chemistry and relatively low price[20–23]. Recent findings show that the number of carbon layers, oxidation states, and surface groups are closely related to the SERS properties of carbon substrates[24,25]. In this sense, understanding the Raman enhancement mechanism on different carbon allotropes is crucial for both the development of advanced carbon-based SERS sensors and the study of the structure-property relationship of carbon materials.

Unlike graphene and carbon nanotubes, which are composed of $sp^2$-hybridized carbon atoms, diamond that contains $sp^3$-hybridized carbon atoms, and $sp$ hybridized acetylene, graphdiyne (GDY) emerges as a two-dimensional (2D) carbon allotrope that simultaneously contains both $sp^2$ and $sp$ hybridized carbon atoms[26–34]. The highly conjugated structure enables GDY to show properties different from other carbon allotropes, such as an adjustable band structure,

[1]Key Laboratory of Consumer Product Quality Safety Inspection and Risk Assessment for State Market Regulation, Chinese Academy of Inspection and Quarantine, Beijing 100176, P. R. China. [2]School of Physics and Physical Engineering, Qufu Normal University, Qufu 273165, P. R. China. [3]School of Materials and Chemistry, China Jiliang University, Hangzhou 310018, P. R. China. [4]School of Chemistry, Beijing Normal University, Beijing 100875, P. R. China. ✉e-mail: xiguangcheng@caiq.org.cn; lqmao@bnu.edu.cn

regularly distributed pores, and abundant chemical states. Different from graphene, GDY is an intrinsic semiconductor[35]. This uniqueness brings opportunities for various applications, such as catalysis, lithium storage, dye-sensitized solar cells, biomedicine, sensors, and oil/water separation[36–39]. For instance, Yan et al. demonstrated that GDY oxide has an ultrafast response to humidity[36]. Gao et al. reported that GDY shows strong electrogenerated chemiluminescence (ECL)[39]. Recently, we reported a surfactant-based microemulsion method for synthesizing GDY hollow microspheres and reported their pollutant adsorption and Raman sensing properties[40]. Considering the possible interference of surfactants with various physicochemical processes, a surfactant-free synthesis and the underlying mechanism of Raman enhancement need to be further explored.

In this study, GDY hierarchical hollow microspheres (HHMSs) with self-supporting structures and ultrahigh specific surface areas were synthesized through a surfactant-free liquid–liquid interface-induced growth method. Furthermore, we find that the semiconducting property of GDY enables its application as a highly sensitive SERS substrate. The Raman enhancement factor (EF) and detection limit of the GDY HHMSs are up to $3.7 \times 10^7$ and $1.0 \times 10^{-12}$ M for rhodamine 6 G (R6G), respectively. The sensing properties of GDY originate from the strong interfacial interactions within the GDY-molecule system.

## Results and discussion
### Synthesis and characterization of GDY HHMSs
GDY HHMSs were prepared by an improved Glaser-Hay coupling reaction with hexaethynylbenzene (HEB) as the monomer[32]. As shown in Fig. 1a, the coupling reaction was designed to be carried out at the liquid–liquid interface of the aqueous and organic phases. The aqueous solution containing $Cu^{2+}$ ions was poured into the $CHCl_3$ solution containing HEB, stirred vigorously for 5 min and then kept at rest to obtain a layered liquid (Fig. 1b)–without using surfactants. After standing in a dark environment at room temperature for 4 days, a layer of dark-brown material grew on the two-phase interface (Fig. 1b). The products were centrifuged and washed with dilute hydrochloric acid, deionized water, and absolute ethanol. Finally, the product was dried in a vacuum drying oven at 50 °C for 3 h.

GDY has a unique 2D planar network structure, which is formed by inserting diacetylenic linkages between two benzene rings in the graphene structure (Supplementary Fig. 1). The product prepared by the interface coupling reaction shows a strong diffraction peak at 21.3° in the X-ray diffraction (XRD) pattern (Supplementary Fig. 2), which is a characteristic peak of the (002) reflection of GDY[32]. The Raman scattering peaks at 1936.4 cm$^{-1}$ and 2172.9 cm$^{-1}$ shown in Fig. 1c were attributed to the conjugated diyne linkage ($-C \equiv C-C \equiv C-$), which proves the existence of $sp$-hybridized carbon atoms[41]. Meanwhile, the peaks at 1396.9 cm$^{-1}$ and 1577.8 cm$^{-1}$ were the D-band and G-band, respectively. Among them, the D-band represents the disorder caused by various defects, while the G-band represents the order of the lattice. The relative strength of the ID/IG is 0.49, indicating a low level of disorder in GDY.

The elementary composition and bonding structure were investigated by X-ray photoelectron spectroscopy (XPS). The carbon-to-oxygen ratio in GDY, as calculated from the survey spectrum, is 8.1 (Supplementary Fig. 3), which is identical to the result obtained by energy-dispersive spectrometry (EDS, Supplementary Fig. 4). The binding energy at 284.8 eV is the C1s orbital, which was fitted into four subpeaks, corresponding to the C=C bond (284.5 eV), C≡C bond (285.2 eV), C–O bond (286.5 eV) and C=O bond (288.9 eV) (Fig. 1d)[42]. The binding energy at 532.3 eV is the C1s orbital, which was fitted into two subpeaks (533.1 eV and 531.8 eV), corresponding to surface functional groups, such as hydroxyl and carbonyl groups (Supplementary Fig. 5). The appearance of oxygen-containing groups originates from the oxidation of some terminal alkynes[32]. UV–Vis absorption was employed to investigate the optical properties of the as-synthesized

GDY. Compared with the HEB monomer, a significant bathochromic shift was observed (Supplementary Fig. 6), suggesting greatly enhanced electron delocalization with the extended conjugated π-system[43].

We systematically characterized the microstructures of the GDY HHMSs prepared by the interface coupling reaction. Scanning electron microscopy (SEM) images show that the GDY particles present a spherical structure composed of a large number of nanosheets (Fig. 1e and Supplementary Fig. 7a). A magnified image reveals that the thickness of the nanosheets is approximately 2.5 nm (Supplementary Fig. 7b, c), which is highly consistent with the atomic force microscopy (AFM) characterization results (Supplementary Fig. 8). Some of the broken microspheres showed empty interiors (Supplementary Fig. 9). Transmission electron microscopy (TEM) images further demonstrate that these microspheres are hierarchical hollow spheres composed of ultrathin nanosheets (Fig. 1f, g and Supplementary Fig. 10). Selected area electron diffraction (SAED) of individual nanosheets exhibits a regular hexagonal lattice pattern (Supplementary Fig. 11), indicating that these GDY HHMSs have high crystallinity. Furthermore, a well-defined periodic pore structure can be clearly seen in a spherical aberration corrected high-resolution TEM (HRTEM) image (Fig. 1h) and the corresponding FFT pattern (inset in Fig. 1g), demonstrating the successful synthesis of high-quality GDY. The specific surface area of the GDY HHMSs is up to 1182 m$^2$ g$^{-1}$ (Supplementary Fig. 12).

### Formation mechanism of GDY HHMSs
We then explored the formation mechanism of these GDY HHMSs in the absence of surfactants. The comparative experimental results show that when the two liquids were mixed very gently (without stirring), dark-brown products were also formed at the interface of the two-phase liquids after 4 days. SEM and TEM images show that these reaction products are hierarchical nanoplates (HNPs) composed of densely arranged nanosheets (Supplementary Fig. 13), no HHMS structures were found. The thickness of the nanosheets is approximately 2.5 nm, essentially the same thickness that makes up the HHMSs. The comparative experimental results suggest that stirring behaviour is an important influencing factor for the formation of GDY HHMSs. It is reasonable to think that after vigorous stirring, two immiscible solutions collide to form a microemulsion layer (MEL) at the interface of the two-phase liquids. The existence of these spherical microemulsion droplets was confirmed by cryo-electron microscopy (Supplementary Fig. 14). Interestingly, we note that the inner wall of these HHMSs is relatively smooth, while their outer surfaces are composed of nanosheets (Supplementary Fig. 15).

Considering that GDY is hydrophilic[40], we speculate that these spherical microemulsions belong to the oil-in-water (O/W) type, which satisfies this requirement of the characteristics of nanosheets growing from the inside out. EDS characterization revealed that the microemulsion belongs to an oil ($CHCl_3$) in water (O/W) configuration (Supplementary Fig. 16). Using the O/W microemulsion droplets as templates, the HEB monomers inside the $CHCl_3$ droplets are continuously catalysed by the $Cu^{2+}$ ions outside to form this HHMS structure. The continuous monitoring of the growth process of the GDY HHMSs also provides clear evidence for our above inference (Fig. 2a–d). Specifically, after one day of reaction, the obtained product was GDY hollow spheres, and no nanosheets were found on the surface (Fig. 2a). After 2 days, the surface of the hollow spheres was covered with a dense layer of small nanosheets (Fig. 2b, c). Eventually, when the time was extended to 4 days, GDY HHMSs formed (Fig. 2d). The proposed interface-induced growth mechanism is summarized in Fig. 2e.

### SERS properties of GDY HHMSs
The SERS properties of the GDY HHMSs were systematically studied. In the Raman tests, R6G was used as a probe with an excitation wavelength of 532 nm. We found that the GDY-HHMS substrate

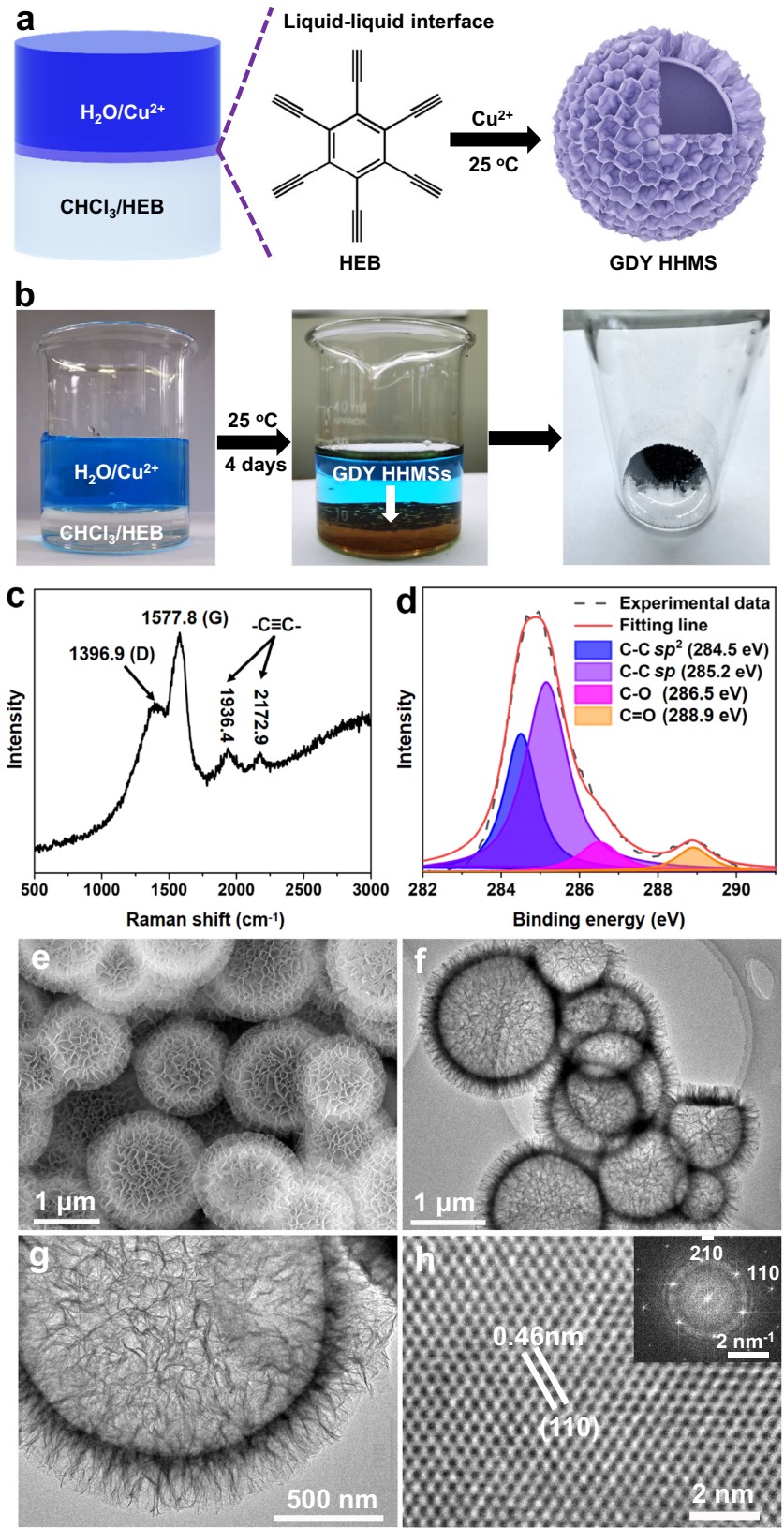

**Fig. 1 | Synthesis and structure characterizations of GDY HHMSs. a** Synthesis diagram. **b** Photos of the liquid–liquid interface before and after the reaction. **c** Raman spectrum of the dry GDY HHMSs. **d** XPS spectrum of C 1 s. **e** SEM image. **f**, **g** Low- and high-magnification TEM images. **h** HRTEM image and corresponding FFT pattern. Source data are provided as a Source Data file.

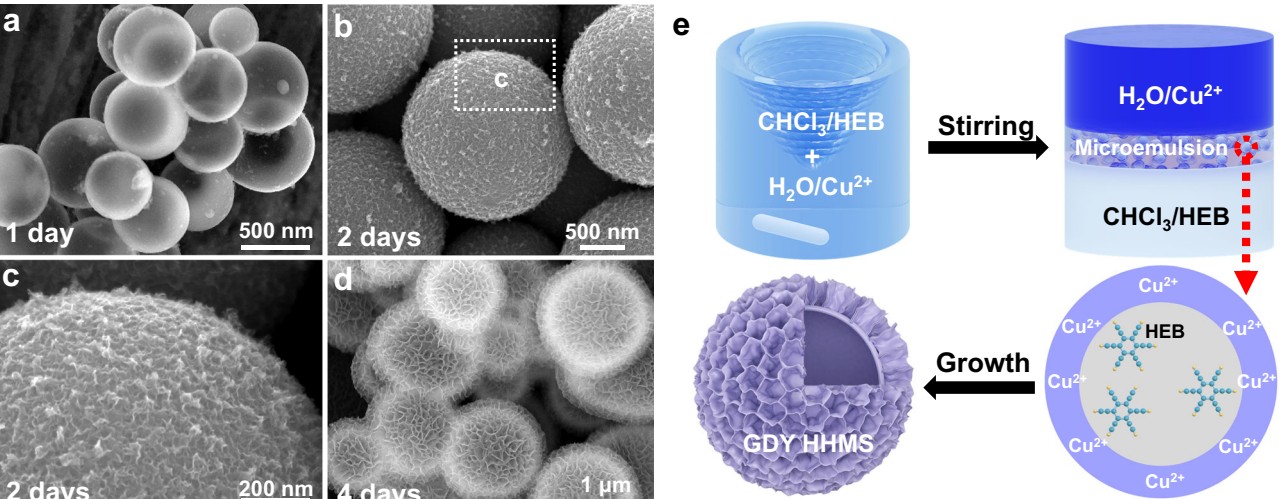

**Fig. 2 | Investigation of the formation mechanism of GDY HHMSs. a** After one day, the product was hollow GDY microspheres. **b**, **c** After 2 days, the surface of the hollow spheres was covered with a dense layer of small nanosheets. **d** After 4 days, the product was GDY HHMSs. **e** Formation diagram of the proposed interface-induced growth mechanism for GDY HHMSs.

(Supplementary Fig. 17) shows a strong SERS effect for R6G molecules (Fig. 3a). Four typical Raman scattering peaks of R6G, $R_1$ (612 cm$^{-1}$), $R_2$ (772 cm$^{-1}$), $R_3$ (1362 cm$^{-1}$), and $R_4$ (1650 cm$^{-1}$), were clearly observed. Among them, $R_1$ and $R_2$ refer to the in-plane and out-of-plane bending vibrations of the C and H atoms of the xanthenes skeleton, respectively; $R_3$ and $R_4$ are attributed to the C−C stretching vibrations of the aromatic nucleus[44], which is highly consistent with the Raman spectrum of R6G powders (Supplementary Fig. 18). Parallel experiments were carried out to distinguish whether the glass wafer played a role in the SERS effect since GDY nanosheets covered it. The results indicated that no SERS signals were detected when a pristine glass wafer was used (Supplementary Fig. 19), which expressly excludes the action of the glass wafer in the SERS behaviour. GDY nanosheets present a good response to R6G molecules in a large concentration range, and the limit of detection can even reach as low as $1 \times 10^{-12}$ M (Supplementary Fig. 20).

Benefiting from the unique ordered 2D nanopore structure of GDY, the signal uniformity of GDY HHMSs is high. For 20 randomly detected sites in the GDY substrate, the results indicated that the peak intensities of the 20 Raman spectra of $1 \times 10^{-9}$ M R6G are highly consistent (Fig. 3b). To evaluate the signal uniformity of this substrate more accurately, we used Raman mapping technology to scan 5000 sites in 2 cm². Statistical analysis of the $R_1$ intensities shows that the relative standard deviation (RSD) is only 4.3% (Fig. 3c). Signal repeatability tests were conducted on the other two substrates of the same batch, and the results showed that their RSD values were similar to the RSD value obtained from the first substrate (Supplementary Fig. 21). The uniform SERS-mapping image also clearly proves the low RSD (Supplementary Fig. 22). As a comparison, the commercial Au nanoparticle-based SERS substrate (Supplementary Fig. 23a) has a much higher RSD value of 19.5% (Supplementary Fig. 23b), showing the obvious signal advantage of GDY.

The R6G signal intensity recorded on pristine glass and the GDY HHMSs was used to calculate the Raman EF of the GDY HHMSs (Fig. 3d). The signal intensities of $R_1$ and $R_2$ at three concentrations ($10^{-9}$, $10^{-10}$, and $10^{-11}$ M) were detected. To ensure the accuracy of the results, the intensity of each peak at each concentration was calculated on average from 20 measured sites on the substrates. For $R_1$, a series of tremendous EFs were obtained at each concentration. The EF for $R_1$ reached $3.7 \times 10^7$ at $10^{-11}$ M, which is approximately 1000 times higher than that of graphene nanosheets ($\approx 10^4$ for $10^{-5}$ M R6G)[20,24]. For $R_2$, the

value also reached the $10^7$ level. It was noted that the EF value showed a gradually escalating trend in the low concentration region, which can be attributed to the fact that the lower the concentration is, the closer the molecular adsorption is to monolayer adsorption[45]. Considering that the 532 nm excitation (2.33 eV) is very close to the energy gap of R6G (2.3 eV), to eliminate the contribution of molecular resonance, we replaced the 532 nm excitation with 633 nm excitation (1.95 eV). The results showed that under 633 nm excitation, R6G molecules obtained an EF of $2.7 \times 10^6$ on these GDY HHMSs. Moreover, the GDY HHMSs also show SERS sensing responses to high-risk health hazards such as bisphenol A (BPA) and dichlorophenol (2,4-DCP) promulgated by the WHO (Fig. 3e, f), indicating their potential application prospects.

**Mechanistic origin of the SERS activity**

We investigated the mechanistic origin of Raman scattering enhancement of GDY. Different from conductive graphene, GDY is an intrinsic semiconductor[35]. For the GDY HHMSs, the energy gap is approximately 1.71 eV according to the UV–Vis absorption spectrum (Fig. 4a). Combining the results of the Mott-Schottky curve (Supplementary Fig. 24) and the XPS valence-band spectrum (Supplementary Fig. 25), the calculated VB and CB levels (vacuum) of the GDY HHMSs are −5.32 and −3.61 eV, respectively. The HOMO and LUMO energy levels of R6G molecules are −5.70 and −3.40 eV, respectively. Combining the above data, a chemical enhancement (CM) dominated by photoinduced ICT is proposed (Fig. 4b). From the point of view of energy matching, it can be expected that contributions from several types of thermodynamically feasible ICT resonance may be related to the overall CM enhancement in our GDY-R6G system at an excitation of 532 nm (2.33 eV), including the molecular resonance of R6G ($\mu_{mol}$, 2.3 eV), the exciton resonance of GDY ($\mu_{ex}$, 1.71 eV), and the photon-induced charge transfer resonance ($\mu_{ICT-1}$: 2.09 eV and $\mu_{ICT-2}$: 1.92 eV) together with the ground-state charge transfer resonance ($\mu_{GSCT}$, 0.38 eV) from the matched energy level between GDY and R6G molecules.

We shall note that the Raman EF of crystalline graphene nanosheets with a thickness of 3.5 nm is only $2.1 \times 10^4$ under the same measurement conditions (Supplementary Fig. 26), which is consistent with previous reports[24,44]. To further clarify the underlying mechanism for the higher SERS activity (1000 times) of GDY relative to graphene, we carefully evaluated the interaction of R6G molecules with GDY and graphene by using spectroscopic analysis. The significant interface interaction between GDY and R6G is clearly observed using UV–Vis

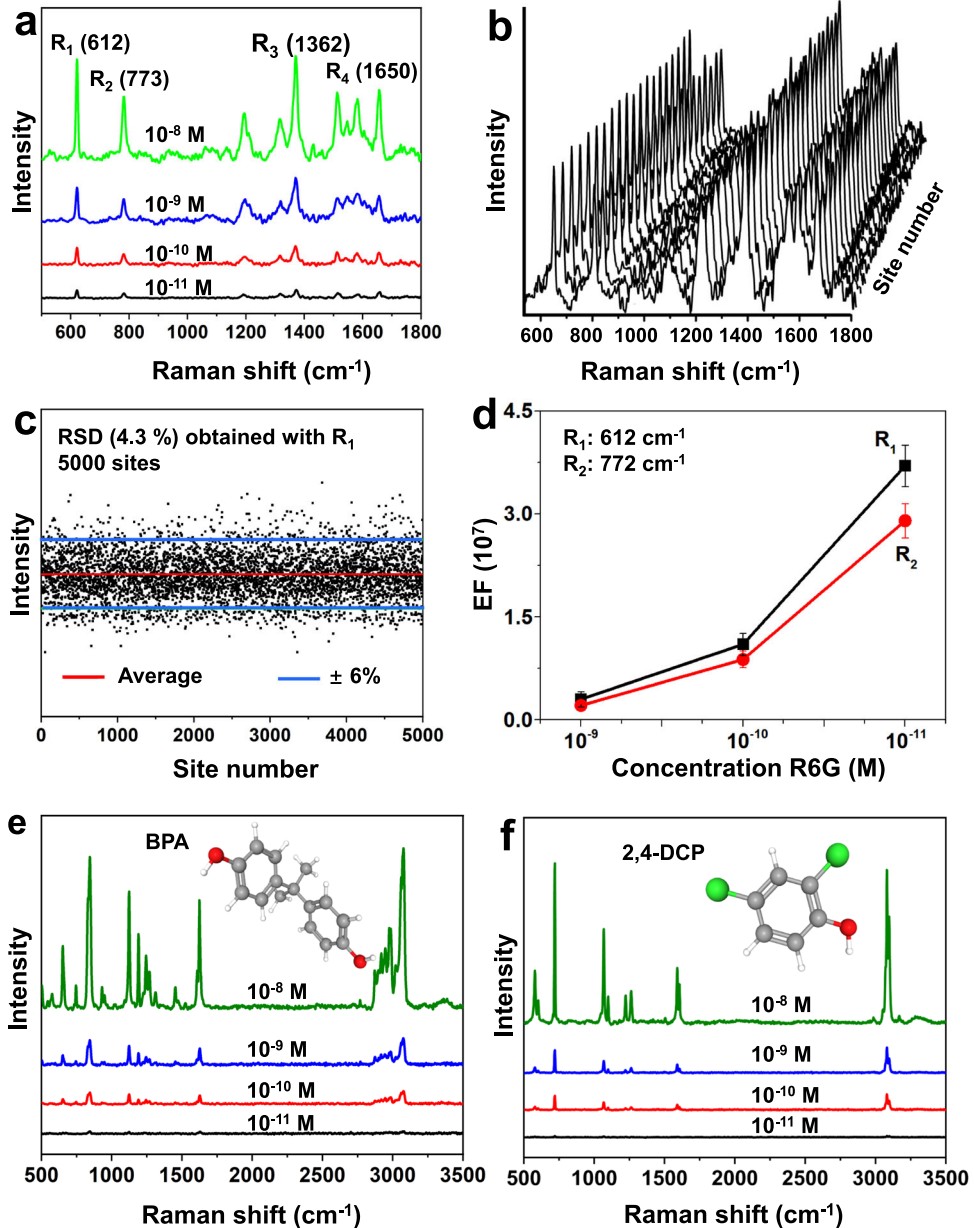

**Fig. 3 | SERS properties of GDY HHMSs. a** SERS spectra of R6G samples recorded on dry GDY HHMS substrates. **b** SERS signals collected from 20 randomly selected sites on GDY HHMSs. **c** RSD of SERS signal intensities at 5000 sites. **d** Raman EFs with R1 and R2 at different concentrations. The error bars are based on the standard deviations of 10 measurements at each concentration. **e** SERS spectra of BPA. **f** SERS spectrum of 2,4-DCP. Excitation wavelength: 532 nm, laser power: 0.7 mW, integration time: 2 s for $10^{-8}$ M, 5 s for $10^{-9}$ M R6G, 15 s for $10^{-10}$ M, and 40 s for $10^{-11}$ M (BPA, 2,4-DCP). The fluorescent background of the probe molecule was removed. Source data are provided as a Source Data file.

spectroscopy (Fig. 4c), as evidenced by a changed absorption spectrum when R6G molecules are chemisorbed at the surface of GDY. A new absorption peak appears at approximately 461 nm, accompanied by a noticeable colour change in the GDY/R6G mixture (see inset of Fig. 4c), while such changes are not as prominent in the graphene/R6G mixture. The newly formed absorption peak indicates significant ICT between GDY and R6G. Additional ICT evidence is obtained by examining the photoluminescence (PL) spectra of R6G. As observed in Fig. 4d, the PL emission peak of R6G (557 nm), which is attributed to the radiative transition of excited singlet state electrons to the ground state, is quenched by approximately 79% when GDY is added. Considering that the specific surface areas of graphene (985 $m^2\,g^{-1}$, Supplementary Fig. 27) and GDY (1182 $m^2\,g^{-1}$) are almost on the same order of magnitude, it is certain that interfacial interactions are the intrinsic

factor responsible for the significantly different SERS properties of GDY and graphene.

To further clarify the underlying mechanism for the higher SERS activity of GDY relative to graphene, we carefully evaluated the interaction between R6G molecules and GDY by using first-principles simulations. A $2 \times 2 \times 1$ supercell of GDY was used to calculate the adsorption energy and charge interaction with the R6G molecule. First, the energy level of the frontier $p_z$ orbital is more similar in the GDY co-adsorption system than in graphene because the orbitals of $sp$ hybridized carbon atoms in GDY are more local and present more peaks in the projected density of state (PDOS) than those of total $sp^2$ hybridized carbon atoms in graphene. The first peak above the Fermi level in the PDOS of GDY and R6G is at 0.41 eV and 0.36 eV, respectively, and the first peak below the Fermi level in the PDOS is at −1.48 eV and −1.38 eV,

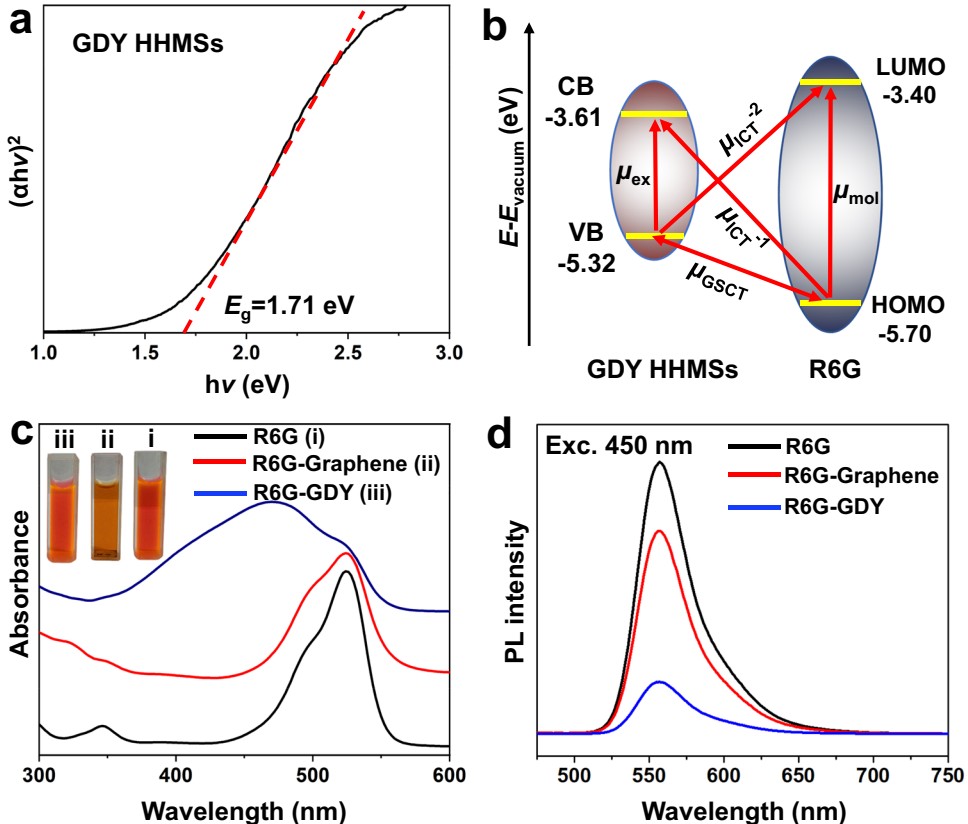

**Fig. 4 | ICT effect between GDY and R6G. a** The Tauc plot curve shows a direct optical bandgap at 1.71 eV for GDY HHMSs. **b** Band energy alignment diagram of the charge-transfer pathways in GDY HHMSs. ICT: interfacial charge transfer, GSCT: ground-state charge transfer resonance. **c** Absorption spectra of pristine R6G solution (5 mg L$^{-1}$) and R6G solution (5 mg L$^{-1}$) mixed with GDY HHMSs (10 mg L$^{-1}$) and graphene nanosheets (10 mg L$^{-1}$). The inset shows the colour of the respective solutions. **d** Photoluminescence spectra of pristine R6G solution (5 mg L$^{-1}$) and R6G solution (5 mg L$^{-1}$) mixed with GDY HHMSs (10 mg L$^{-1}$) and graphene nanosheets (10 mg L$^{-1}$). Source data are provided as a Source Data file.

respectively (Fig. 5a). The difference in the frontier orbital energy level is less than 0.10 eV, which is much smaller than that in the graphene co-adsorption system (>0.71 eV, Fig. 5b). Second, the first-principles calculation demonstrates that the adsorption energy of the R6G molecule on GDY is −2.24 eV, and the Bader charge transfers 1.06 e from the R6G molecule to GDY. Moreover, the injected electrons are distributed on the whole surface of GDY, and the charge accumulation (marked as yellow) is remarkably located at the surface of GDY (Fig. 5c), which can further induce a strong ICT effect between R6G and GDY. Although 0.72 e is transferred from R6G to graphene and the adsorption energy is 2.09 eV, charge accumulation and depletion mainly occur inside R6G, and there is little charge accumulation on the surface of graphene (Fig. 5d); herein, the SERS effect of graphene is much weaker than that of GDY, although graphene possesses excellent fluorescence quenching capability. Finally, when GDY adsorbs the R6G molecule, the PDOS shows that the magnetic moment of the R6G molecule disappears from 1.0 μB per molecule (Fig. 5a); conversely, the magnetic moment of the R6G molecule does not change after adsorption on graphene (Fig. 5d). Notably, the unpaired electron of R6G transfers to GDY, resulting in a pronounced ICT effect, which is crucial for SERS. In summary, the multiple peaks in PDOS reveal that *sp* hybridized carbon atoms in GDY play a crucial role in enhancing the ICT between GDY and R6G because the *sp* carbon atoms provide more local flat bands to match the molecular orbital of R6G (Fig. 5a), but for graphene, the total sp$^2$ hybridized *p*$_z$ orbital of carbon atoms is delocalized, and it is difficult to form an ICT effect with R6G (Fig. 5d). Therefore, GDY shows a much stronger SERS effect than graphene.

We also explored possible chemical enhancement mechanisms in the GDY/2,4-DCP and GDY-BPA systems via DFT calculations. The interaction electrons distributed on the surface of GDY/2,4-DCP and GDY/BPA are shown in Supplementary Fig. 28, which indicates the strong chemical interactions between the GDY substrate and probe molecules. For 2,4-DCP, the calculated HOMO and LUMO energy levels are −5.73 eV and −1.90 eV, respectively. At an excitation of 532 nm (2.33 eV), it can be expected that contributions from three types of thermodynamically feasible ICT resonance may be related to the overall CM enhancement in the GDY-2,4DCP system, including the exciton resonance of GDY (μ$_{ex}$, 1.71 eV), the photon-induced charge transfer resonance (μ$_{ICT}$: 2.12 eV), and the ground-state charge transfer resonance (μ$_{GSCT}$, 0.41 eV) from the matched energy level between GDY and 2,4-DCP molecules (Supplementary Fig. 29). For BPA, the HOMO and LUMO energy levels are -5.19 eV and −1.35 eV, respectively (Supplementary Fig. 30). Similar to 2,4-DCP, three types of thermodynamically feasible ICT resonance were also found in the GDY-BPA system, including the exciton resonance of GDY (μ$_{ex}$, 1.71 eV), the photon-induced charge transfer resonance (μ$_{ICT}$: 1.58 eV) and the ground-state charge transfer resonance (μ$_{GSCT}$, 0.13 eV). The calculated results clearly reveal the same conclusion that the ICT effect plays a crucial role in the SERS mechanism.

Interestingly, under the same experimental conditions, when we switched the excitation light to 785 nm (1.58 eV), there was no order of magnitude change in the signal intensity of BPA compared to 532 nm excitation. Supplementary Fig. 31 shows the Raman spectra of BPA at a concentration of 1 × 10$^{-8}$ M under 532 nm and 785 nm excitation. The

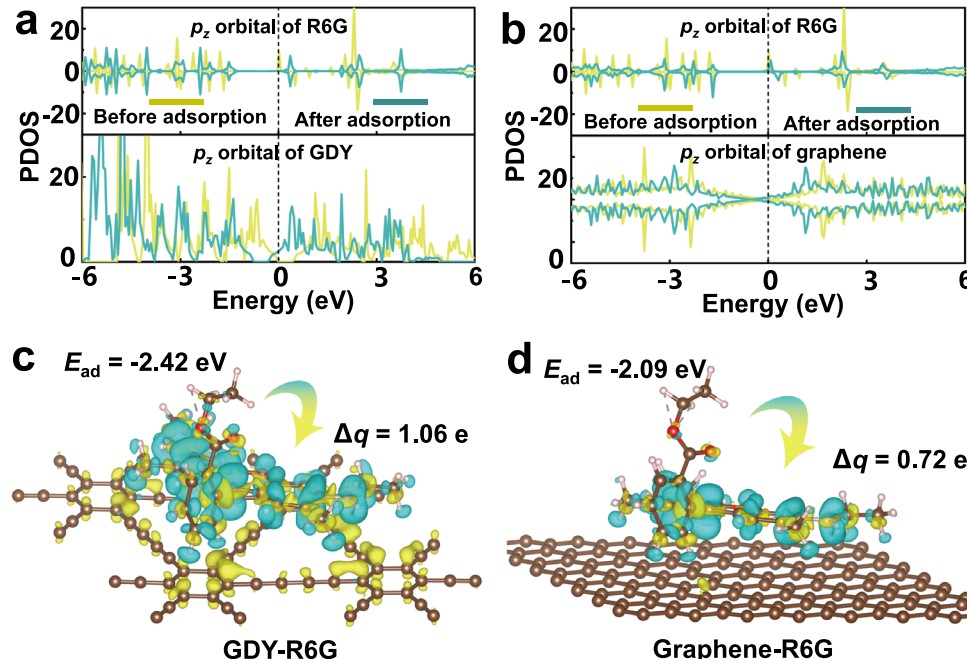

**Fig. 5 | Interfacial interaction between R6G molecules and GDY.** Projected density of states of R6G molecules adsorbing on (**a**) GDY and (**b**) graphene. Charge density difference of R6G adsorbing on (**c**) GDY and (**d**) graphene, where the yellow and cyan colour mark the regions of charge accumulation and depletion, respectively. The isosurface was set to 0.001 e Å$^{-3}$.

Raman peak intensity under 532 nm excitation is stronger than that under 785 nm excitation, but there is no difference in magnitude, which indicates that there is no significant difference in the enhancement effect generated by the two types of excitation. To further clarify this point, we further explored the Raman signal enhancement effects generated by two types of excitation at $1 \times 10^{-10}$ M. The experimental results show that under the same laser power (0.7 mW) and integration time (40 s), the Raman enhancement effects generated by the two excitations are almost the same (Supplementary Fig. 32). The calculated Raman EFs are $2.4 \times 10^6$ (532 nm) and $1.8 \times 10^6$ (785 nm) at $1 \times 10^{-10}$ M, respectively (for the calculation process, see Methods). The possible reasons for this similarity are that although the energy of 785 nm is consistent with that of the required photon energy (1.58 eV) for ICT between GDY HHMS and BPA, the energy of 785 nm excitation cannot initiate the excitation resonance of GDY ($\mu_{ex}$, 1.71 eV). That is, compared to 532 nm excitation, there are only two forms of thermodynamically feasible resonance under 785 nm excitation (Supplementary Fig. 33). Therefore, overall, there is no significant difference in the enhancement effect between the two excitations.

We also calculated the Raman EFs of 2,4-DCP under 532 nm and 785 nm excitation. The calculation results indicate that under 532 nm excitation, the Raman EF obtained is $4.3 \times 10^6$ (for the calculation, see Methods). However, when 532 nm excitation was replaced with 785 nm excitation, no 2,4-DCP signals were found, except for the fluorescent background (Supplementary Fig. 34). We assume that this is because the energy excited by 785 nm (1.58 eV) is too low to drive the exciton resonance of GDY ($\mu_{ex}$, 1.71 eV) or initiate interface ICT resonance ($\mu_{ICT}$, 2.12 eV). Therefore, under 785 nm excitation, only one weak $\mu_{GSCT}$ resonance (0.41 eV) exists, which cannot produce a distinguishable Raman spectrum.

In summary, GDY HHMSs were successfully synthesized through a surfactant-free growth method utilizing liquid–liquid interfaces. SERS activity of up to $3.7 \times 10^7$ was observed in this intrinsic semiconductor carbon materials. The potential mechanism behind the SERS phenomenon was studied through DFT simulations. The SERS effect of GDY is proposed to be dominated by the strong interfacial interaction.

The current work provides experimental and theoretical proofs for understanding the difference in the SERS properties of GDY and graphene. Considering the abundance of GDY raw materials with high chemical stability (Supplementary Figs. 35 and 36), GDY is likely to become a practically relevant nonmetallic SERS substrate. Moreover, the energy band structure of GDY can be adjusted by means of element doping and surface modification[29,30], which provides the possibility for selective molecular detection[46–50]. This has been preliminarily proven on other semiconductor SERS substrates, such as metal-organic-framework compounds and oxygen-deficient nanostructures[7,51]. Our findings will guide further development in the structural design and fabrication of high-performance SERS substrates, in addition to fuelling the exploration of GDY nanostructures in diverse applications.

## Methods
### Synthesis of GDY HHMSs
In a typical synthesis, 30 mg of hexakis[(trimethylsilyl)ethynyl]benzene (HEB) was dissolved in 10 mL of CHCl$_3$, to which 10 μL of tetrabutylammonium fluoride dissolved in 1 M THF was added. After standing for 5 min, 25 mL of aqueous solution containing 0.15 g of pyridine and 25 mg of copper acetate was added to the above solution. The obtained mixed solution was stirred vigorously (300 rpm) by a magnetic stirrer for 5 min. The reaction solution was left to rest at room temperature in the dark for 4 days. After the reaction, the obtained black product was washed three times with deionized water, dilute hydrochloric acid (1 M), and absolute ethanol and dried in a vacuum oven at 50 °C for 3 h. Compared to previous methods, the current experiment does not require additional inert gas protection because the reaction occurs at the liquid-liquid interface.

### Synthesis of GDY HNPs
In a typical synthesis, 30 mg of hexakis[(trimethylsilyl)ethynyl]benzene (HEB) was dissolved in 10 mL of CHCl$_3$, to which 10 μL of tetrabutylammonium fluoride dissolved in 1 M THF was added. After standing for 5 min, 25 mL of aqueous solution containing 0.15 g of pyridine and 25 mg of copper acetate was added to the above solution.

The obtained mixed solution was left to rest at room temperature in the dark for 4 days. After the reaction, the obtained black product was washed three times with deionized water, dilute hydrochloric acid (1 M), and absolute ethanol and dried in a vacuum oven at 50 °C for 3 h.

## SERS experiments

To study the SERS properties of the as-synthesized GDY HHMSs, a confocal micro-Raman spectrometer (Renishaw-inVia Qontor) was used. When R6G is used as the probe molecule, the excitation wavelength used is 532 nm. When using BPA and 2,4-DCP as probe molecules, in addition to 532 nm excitation, 785 nm excitation was also used. In all SERS measurements, unless specifically stated, the laser power is 0.7 mW, and the specification of the objective is 50x (L). The laser comes with the instrument and has not undergone further monochromation. A series of standard solutions of R6G, BPA, and 2,4-DCP with concentrations of $10^{-8}$–$10^{-12}$ M were used as the probe molecules. To improve the signal reproducibility and uniformity, 30 mg of GDY HHMSs was dispersed into a probe solution (30 mL) to be measured. After stirring for 3 min, the GDY HHMSs were removed by centrifugation and dried in air at room temperature for 5 min. In all SERS tests, the laser beam was perpendicular to the top of the sample to be tested with a resultant beam spot diameter of 5 μm. During the operation of the laser, the room was kept dark. The fluorescent background of the probe molecule was removed by the software that comes with the instrument.

## Calculation of Raman EFs

The Raman EF of the GDY HHMSs was calculated according to the equation[52]

$$EF = \frac{I_{SERS}}{I_{bulk}} \frac{N_{bulk}}{N_{SERS}} \qquad (1)$$

where $I_{SERS}$ and $I_{bulk}$ are the Raman intensities in the SERS experiments and of the bulk dye molecules, respectively. $N_{SERS}$ and $N_{bulk}$ are the amounts of molecules involved in the SERS experiments and in the bulk Raman measurements, respectively.

## Calculation of R6G Raman EF

We used the R6G peak at 612 cm$^{-1}$ to estimate the EFs. In the experiments, the excitation wavelength was 532 nm, the laser power was 0.7 mW, and the integration time was 40 s. The peak intensity at 612 cm$^{-1}$ of R6G/GDY ($1.0 \times 10^{-11}$ M) was 31.5 counts with a 40 s acquisition time, and that of bulk R6G was 421.5 counts with a 0.5 s acquisition time. With normalization with respect to the acquisition time, the Raman intensity ratio was estimated to be $I_{SERS}/I_{bulk} = (31.5/40)/(421.5/0.5) = 9.3 \times 10^{-4}$. As shown in Supplementary Fig. 37, in the experiments, 10 mg of GDY HHMSs was dispersed in 3 mL of anhydrous ethanol and ultrasound for 2 min. The obtained uniform GDY suspension droplets were added into a glass circular groove with an area of approximately 0.8 m$^2$. After the ethanol volatilized, the remaining suspension was dripped into the groove until all the suspension was used up. Finally, a layer of GDY substrate was formed in the groove, with a thickness of approximately 0.1 mm. A 10 mL volume of the dye solution was drop-cast on the SERS material with a substrate area of ≈0.8 cm$^2$, followed by a gentle drying process. $N_{SERS}$ was written as $N_{SERS} = c V N_A (A_1/A_{sub})$, where $c$ is the dye concentration, $V$ is the dye droplet volume, $N_A$ is the Avogadro constant, $A_1$ is the laser spot area, and $A_{sub}$ is the substrate area. To obtain $N_{bulk}$, a high-concentration (0.2 M) R6G solution was drop-cast onto bare glass. The bulk R6G crystals (thickness is larger than the laser penetration depth) were formed when the solution was dried out. Hence, we could use the density of R6G to calculate the number of molecules. $N_{bulk}$ is written as $N_{bulk} = d h N_A (A_1/M)$, where $d$ is the density of bulk dye (1.15 g cm$^{-3}$ for R6G), $h = 21$ μm is the laser penetration depth[53], and $M$ is the molar

mass of dye (479.01 g mol$^{-1}$ for R6G). Taking all the above-mentioned factors into account, the EF in the R6G/GDY measurements was derived as:

$$EF = \frac{I_{SERS}}{I_{bulk}} \times \frac{dhA_{sub}}{cVM} = 9.3 \times 10^{-4} \times \frac{1.15 \times 21 \times 10^{-4} \times 0.80}{1 \times 10^{-11} \times 10 \times 10^{-6} \times 479.01} = 3.7 \times 10^{7}$$

$$(2)$$

## Calculation of BPA Raman EF

The calculation method of BPA EFs is consistent with the calculation method of R6G. In the experiments, the excitation wavelength was 532 nm, the laser power was 0.7 mW, and the integration time was 40 s. Here, we used the BPA peak at 846.7 cm$^{-1}$ to estimate the EFs. The peak intensity at 846.7 cm$^{-1}$ of BPA/GDY ($1.0 \times 10^{-10}$ M) is 4.75 counts with a 40 s acquisition time and that of bulk BPA is 215.1 counts with a 0.5 s acquisition time. The density of bulk BPA is 1.19 g cm$^{-3}$, and the molar mass of BPA is 228.28 g mol$^{-1}$. Taking all the factors into account, the EF in the BPA/GDY measurements was derived as $2.4 \times 10^{6}$. Keeping the conditions unchanged, when only changing the excitation wavelength to 785 nm, the peak intensity at 846.7 cm$^{-1}$ of BPA/GDY ($1.0 \times 10^{-10}$ M) is 3.43 counts with a 40 s acquisition time and that of bulk BPA is 198.3 counts with a 0.5 s acquisition time. The EF in the BPA/GDY measurements (785 nm) was derived as $1.8 \times 10^{6}$.

## Calculation of 2,4-DCP Raman EF

The calculation method of 2,4-DCP EFs is consistent with the calculation method of R6G. In the experiments, the excitation wavelength was 532 nm, the laser power was 0.7 mW, and the integration time was 40 s. Here, we used the 2,4-DCP peak at 865.2 cm$^{-1}$ to estimate the EFs. The peak intensity at 865.2 cm$^{-1}$ of 2,4-DCP/GDY ($1.0 \times 10^{-10}$ M) is 5.16 counts with a 40 s acquisition time and that of bulk 2,4-DCP is 205.2 counts with a 0.5 s acquisition time. The density of bulk 2,4-DCP is 1.38 g cm$^{-3}$, and the molar mass of BPA is 163 g mol$^{-1}$. The EF in the 2,4-DCP/GDY measurements (532 nm) was derived as $4.3 \times 10^{6}$.

## First-principles calculations

In this work, all the first-principles calculations were performed with the Vienna ab initio Simulation Package (VASP)[54,55], the exchange correlation interactions were evaluated using the generalized approximation (GGA) with Perdew-Burke-Ernzerhof (PBE) exchange correlation functional[56,57]. DFT + D3 scheme was considered to describe the van der Walls force and long-range interactions[58]. A vacuum layer with the thickness of 25 Å was set to eliminate the interaction between adjacent periodic sections. The cut-off energy of 600 eV was adopted in all computations, and all structure models were completely relaxed until the force was less than 0.01 eV Å$^{-1}$ and the energy tolerance was less than $10^{-5}$ eV. The Γ-centered Monkhorst-Pack grid k-point sample[59] in Brillouin were set as $3 \times 3 \times 1$ and $6 \times 6 \times 1$ for structure optimization and density of states (DOS) calculations, respectively.

　　The adsorption energy ($E_{ad}$) of R6G molecule adsorption in the surface was define as:

$$E_{ad} = E_{total} - E_{surf} - E_{R6G}$$

where $E_{total}$, $E_{surf}$, and $E_{R6G}$ represents the total energy of co-adsorption system, pristine surface, isolated R6G molecule, respectively. Accordingly, the charge density difference between R6G molecule and surface was computed with follow formula:

$$\Delta\rho = \rho_{total} - \rho_{surf} - \rho_{R6G}$$

where $\rho_{total}$, $\rho_{surf}$, and $\rho_{R6G}$ is the total charge density of co-adsorption system, pristine surface, isolated R6G molecule,

respectively. The qVASP program was used to postprocess raw data of VASP[60].

## Material characterization

GDY HHMSs are characterized by a variety of techniques. XRD patterns of the products are obtained on a Bruker D8 focus X-ray diffractometer by using CuKa radiation ($\lambda = 1.54178$ Å). SEM images and EDS are obtained on a Hitachi S-4800 operated (10 kV). TEM, HRTEM, EDS, and SAED characterizations are performed with a JEOL F200 operated at 200 kV. X-ray Photoelectron Spectroscopy (XPS) experiments are performed in a ESCALab 250Xi using monochromatic Al Kα X-rays at h$\nu$ = 1486.6 eV. Peak positions are internally referenced to the C1s peak (sp$^2$ hybrid carbon binding energy) at 284.1 eV. UV–Vis absorption spectra are recorded with a Shimadzu UV3600-Plus. The specific surface area and pore size are measured in a Micro Tristar II 3020. Raman spectra are recorded from Renishaw-inVia Qontor (wavelength: 532, 633, and 785 nm; laser power: 0.7 mW).

## Data availability

The data that support the findings of this study are available from the corresponding author upon request. Source data are provided with this paper.

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

## Acknowledgements

This work received financial support from the Science Foundation of Chinese Academy of Inspection and Quarantine (No. 2022JK14, G.X.), and National Natural Science Foundation of China (Nos. 21790390, L.M. and 21790391, L.M.).

## Author contributions

G.X. proposed and designed the project. L.Z. prepared materials. L.Z. and J.L. performed UV–Vis, XPS, XRD, SEM, and TEM characterization. L.Z. conducted SERS and EFs measurement. W.Y. performed electronic structure calculations. G.W. and L.M. participated in the enhanced-mechanism discussion. All authors critically evaluated the manuscript.

## Competing interests

The authors declare no competing interests.

## Additional information

**Peer review information** *Nature Communications* thanks Olga Guselnikova and the other, anonymous, reviewer(s) for their con-tribution to the peer review of this work. A peer review file is available.

