## [Peer Review File · Nature Communications]

Surfactant-free interfacial growth of graphdiyne hollow microspheres and the mechanistic origin of their SERS activityEditorial Note: Parts of this Peer Review File have been redacted as indicated to remove third-party material where no permission to publish could be obtained.

Reviewers' comments:

Reviewer #1 (Remarks to the Author):

Authors for the first time demonstrated that one of the carbon allotrope graphdiyne (GDY) shows remarkably Raman signal enhancement activity compared to graphene and other allotropes. The prepared GDY hierarchical hollow microspheres for this and tested R6G and showed the perspective for bisphenol A and dichlorophenol sensing. Also, they tried to explain the enhancement effect by γ interfacial charge transfer due to sp hybridized carbon with R6G. Despite the work is original and reporting first time Raman enhancement on GDY, there is no clarity about the existing/potential advantages compared to existed noble metal materials and recently reported 2d layered materials (MoS₂, Mxene, h-BN) etc.

There are some very critical questions about the concept of this study and details. The major are:

1. Despite the preparation of a high surface area GDY is highly promising in general (catalysis, adsorbent etc), this substrate is not a perfect model for the evaluation of mechanistic aspects of Raman enhancement. This is because for there is a strong dependence of Raman signal intensity on the 2d materials (graphene and other) thickness [J. Am. Chem. Soc. 2018, 140, 8696–8704]. Taking into account the morphology of GDY hierarchical hollow microspheres, this is very difficult to evaluate thickness. This is not clear what was the thickness of SERS substrates with 0.8 cm² and if this thickness is homogeneous over the substrate. Unfortunately, there is no information about the preparation of SERS substrates for EF measurements. The preparation of GDY film has been reported according to the literature [Acc. Chem. Res. 2017, 50, 10, 2470–2478]. So, I suggest confirming these findings on GDY film or providing more detailed characterization of SERS substrate from microspheres.
2. Authors are using R6G and Raman excitation of 532 nm, leading to the surface-enhanced resonant Raman scattering (SERRS) phenomenon, which occurs as excitation laser frequency crosses the frequencies of electronic excited states of the molecule. Although I consider this system not perfectly relevant for mechanistic study because a big part of signal enhancement is coming from this phenomenon, there are extensive examples of this situation. This fact should be underlined in the mechanistic study and SERS and SERRS should not be mixed. The positive part is that authors also showed the SERS activity of GDY

for bisphenol A and dichlorophenol.

3. The experimental details about SERS measurement in Method section and SI are confusing “30 mg of GDY HHMSs was dispersed into a probe solution (30 mL)” and “the dye solution with 10 mL volume was drop-casted on the SERS material with $\sim 0.8 \text{ cm}^2$ substrate area followed by a gentle dry process”. Please, explain, which procedure was used for EF measurement and SERS sensing of other molecules and why these procedures are different

4. What are the general advantages of GDY for SERS compared to common metal-based plasmonic materials providing even higher EF factors?

5. In case of drop deposition of R6G, the formation of the coffee ring effect is common [ChemPhysChem. 2015 16(13):2726-34], how authors addressed this point during EF measurement and comparison with graphene?

Minor questions:

6. How authors showed Raman mapping in 2 cm^2 if the size of the sample was $\sim 0.8 \text{ cm}^2$?

Please, provide this data as a coloured map for the sake of clarity

7. Please, clarify more measurement details, the power 0.7 mW was used with laser spot 5 mm? This is not clear objective $\times 50$ L could give so large spot?

8. Fig. 1b show Raman spectra of GDY, there are 2 peaks at 1932 and 2173 cm^{-1} . Could, you, please, clarify the origin of peak at 1932 cm^{-1} , this is not typical for $\text{C}\equiv\text{C}$

9. Please, add assignation of peaks of bisphenol A and dichlorophenol

10. Please, provide details of graphene origin used in this study

Reviewer #2 (Remarks to the Author):

This article demonstrates spherical porous graphdiyne (GDY) synthesis by using and interface-induced growth mechanism and SERS application with rhodamine 6G (R6G). The spherical porous graphdiyne is a unique structure and homogeneous and it shows high SERS signals by charge transfer. The result looks interesting, but some should be clear for the publication with major revisions in Nature Communications.

1. Page 4, line 8: It is not only noble-metal substrates. Non-noble metals such as aluminum have also been investigated. And, other metals can be candidates for SERS substrates.

Please, check some refs. and add it to the manuscript.

1. Chem. Commun., 2014,50, 3744-3746

2. Appl. Opt. 1995, 34, 4755-4767.

2. Page 4, line 9: Recently, Cu was also reported as SERS substrates.

And various structures should be mentioned because they are also an important factor for SERS enhancement.

Some structural effects of SERS may be very helpful for readers.

please check and add some refs.

1. Small 2022, 18, 2107182

2. The Journal of Physical Chemistry C 124 (43), 23730-23737

3. Journal of Materials Chemistry A 8 (40), 21016-21025

4. RSC advances 10 (14), 8309-8313

3. Page 4, line 11: SERS is not an electromagnetic phenomenon.

It originates from electric field resonance by coherent electron gas. Please double-check and correct it if needed.

4. Page 6, line 6 and 8 (Figure S1a, Figure S1b): No letter "a" and "b" in the Figure S1

5. Page 7, the last two lines: Could you please mention how you corrected the XPS peaks? In general, people use C1s orbital as a ref. But for carbon materials, a different ref. peak should be used to operate deconvolution of C1s.

6. Page 8, line 1: Did you use a Cu TEM grid without any carbon layer for the TEM EDS?? You should mention which TEM grid you used. If you use any holey carbon grid, Figures S4 and 5 shouldn't be the same.

7. Page 8, line3-5: Please conduct O 1s deconvolution as well.

8. Page 8, line12: Scanning electron microscopy (SEM) images shows  show

9. Page 8, line14: It is hard to say that (2.5 nm thickness). Because the resolution of SEM is

generally more than 2.5 nm, although the spec sheet says it is below 2.5 nm. Please check again.

10. Page 11, Chapter "SERS Properties of GDY HHMSs"

How did you count the number of R6G on the SERS substrates? The EF can be easily overestimated due to the underestimated number of molecules. Especially, it is even easier to be underestimated when a material with a high surface is used.

Please describe the details of how you prepared the sample and the reference for the SERS.

11. Page 18-20, Chapter "Method"

There are many typos. Please check if you have done superscripts or subscripts. An entire English check should be done.

Reviewer #3 (Remarks to the Author):

This manuscript reports the graphdiyne (GDY) hierarchical hollow microsphere (HHMS) as a highly sensitive surface-enhanced Raman scattering (SERS) substrate by virtue of its strong chemical enhancement and high specific surface area. The synthesis process is interesting while the quality of the synthesized HHMS is high. The substrate shows excellent SERS performance for several analytes, including R6G, BPA, and 2,4-DCP. Moreover, the authors tried to elucidate its enhancement mechanism by proposing the charge-transfer diagram based on several experimental evidences and density function theory (DFT) calculation. The results are interesting and promising. However, I have several major concerns (see below). I will consider recommending its publication in Nature Communications if the authors can address the following concerns convincingly.

1. The authors claim that their GDY may be the first truly all-carbon semiconducting SERS substrate, which is incorrect. Diamond has been reported as an all-carbon semiconducting SERS substrate [Gao, Ying, et al. "Semiconductor SERS of diamond." *Nanoscale* 10.33 (2018): 15788-15792.]. In addition, although perfect large-area graphene is not a semiconductor while its nanostructures are semiconductors and have been reported for SERS [Das, Ruma,

et al. "Origin of high photoluminescence yield and high SERS sensitivity of nitrogen-doped graphene quantum dots." *Carbon* 160 (2020): 273-286.].

2. From the EDS spectrum in Figure S5, it is evident that Cu which is a well-known SERS active element exists in the synthesized GDY HHMS. The authors should exclude its contribution to the observed SERS performance.
3. In addition to R6G, the authors claim that their substrate also shows excellent SERS performance for BPA and 2,4-DCP. What are the enhancement factors for these two molecules? Is the proposed enhancement mechanism for R6G also applicable for these two molecules while the HOMOs and LUMOs of these two molecules are quite different from that of R6G?
4. In Figure 4c, the authors claim that the newly formed absorption peak at around 461 nm indicates the charge-transfer between GDY and R6G. However, compared with the UV-Vis spectrum of GDY HHMS in Figure S6, it seems that this peak is just the peak of GDY HHMS rather than a newly formed peak.
5. The explanation of the enhancement mechanism is not clear. Based on Figure 5d, the authors claim that the main charge accumulation and depletion only happened inside of R6G when R6G is adsorbed on graphene. How did the authors get this conclusion? Does this conclusion contradict with the excellent fluorescence quenching capability of graphene observed in experiment and in other references.
6. Based on the DFT calculation, the authors claim that the magnetic moment of R6G disappears when it is adsorbed on GDY. What occurs behind this should be clarified. How the magnetic moment of R6G influences the Raman enhancement should be discussed in detail as well.
7. The authors argue that the sp and sp² hybridized carbon atoms are crucial for SERS in their manuscript. A more detailed discussion should be given to justify this.
8. The caption of Figure 5 does not match with the figure panels.

Reviewer #4 (Remarks to the Author):

The authors demonstrated a liquid-liquid interface-induced method to synthesize graphdiyne porous flower like particles and showed SERS capabilities to detect R6G at 10-11

M level. The major concern of the work is that Cu contamination can be noticed via the provided EDX and XPS. Cu nanoparticles also display SERS; thus, therefore it is not clear that only graphdiyne is responsible for the SERS detection limit. Thus, I cannot recommended this paper for publication in Nature Communications.

1. In Figure 1b, the authors have showed the Raman spectra of the as synthesized graphdiyne, other than the D (~1397 cm⁻¹), G (~1577 cm⁻¹), and Y-band (~2173 cm⁻¹), there is a band with relatively high intensity around 1932 cm⁻¹. However, it is noticed that in some literature, the peak around 1932 cm⁻¹ is missing (Appl. Phys. Lett. 113, 021901 2018; DOI: 10.1126/sciadv.aat6378). Could the authors explain the peak around 1932 cm⁻¹?

2. Considering the possible metal or metal oxide contamination, Could the authors also show the Raman spectra starting from 500 cm⁻¹?

3. XPS shown in Figure S4, peaks above 950 eV can be found. Could that be the Cu contamination from the synthesis?

4. The authors used EDX (Figure S5) to show/confirm the C/O ratio. However, EDX is not the ideal characterization technique to quantify light elements like C and O. It would be better to use EELS or XPS, instead of EDX. EDX is good for heavy element, such as Cu. It is clear that Cu is found within the samples based on Figure S5.

5. The authors mentioned that graphdiyne is a direct band gap semiconductor, thus, it is recommended to show the photoluminescence, other than UV-Vis.

6. The morphology of the graphdiyne is the porous flower like particles. How did the authors measure the flake thickness of the particles?

7. In the characterization section, many details are missing. For example, the excitation wavelength of the Raman measurements, the accelerating voltage for SEM and HRTEM, etc.

Reviewer #5 (Remarks to the Author):

The article “Liquid-liquid interface growth of graphdiyne with ultrahigh specific surface area as highly sensitive surface-enhanced Raman scattering substrate” proposed a new micro carbon particle made with graphdiyne, which is composed of hollow core with buddied nanosheets. They emphasized that they conducted the SERS study for the first time as a microparticle made of GDY, and obtained about 1000 times stronger SERS signal than the existing carbon-based particles. In addition, it was theoretically revealed that this SERS enhancement was caused by the ICT mechanism. This is an interesting study and a meaning addition to the literature in related fields with some novel aspects. There are, however, some serious concerns and issues that need to be fully addressed and the manuscript needs to be significantly improved to be considered for publication in Nature Communications. In the current form, this paper fits in a bit more specialized journal.

1. The author stated that Cu²⁺ ion was used as a catalyst, but the EDS result of the product showed a high Cu content (Figure S5) with more than half of carbon in atomic ratio. This seems to prove that this particle is not a pure carbon particle as claimed by the authors, but a composite particle containing an excessive amount of Cu. Also, the color of the product is brown, further showing the existence of copper. If this part is not clearly resolved, the following problems arise.

A. SERS signal that is 1000 times stronger than that of conventional graphene is not produced by pure carbon, but is interpreted as enhancement by Cu surface plasmon.

B. It is not the SERS enhancement by ICT, but the EM enhancement effect. Therefore, many claims and discussions become rather meaningless.

2. The quantitative signal of this particle was claimed by comparing the commercial AuNPs and the SERS signal, but it is not that meaningful nor fair because it was compared with the worst case (Figure S20a).

3. At the EF calculation equation in the supporting information, if I'm not mistaken, the calculation with the numbers and equations given by the authors do not reproduce the given EF of 3.6×10^7 , but suggests EF of 4.4×10^8 . Please recheck the number.

4. The notes say that the peak intensity was acquired with 15 s acquisition time in GDY, but when the Raman intensity ratio was estimated, an equation of $(31.5/40)/(421.5/0.5)$ was used, suggesting that 40s acquisition time was used in the calculation.

5. In the EF calculation equation in the supporting information, r and p are both used as a

notation for the density of bulk dye. It might be beneficial to unify these

6. The authors used a laser penetration depth of 21 mm, possibly brought from the reference. However, the laser penetration depth is different from facility to facility, as they depend on the pinhole size and the objective lens used. Reliable reasoning and experimental values to confirm the laser penetration depth should be provided.

7. Adding the line slice image of AFM in Figure S8 would help readers to confirm the thickness of the nanosheets.

8. The authors projected RSD analysis in Figure 3c. In addition to that, giving a mapping image of SERS signal could also be a good way to show the uniformity of the substrate.

9. A typo is in Figure S8. 'Sinlge' should be changed into 'single'.

10. A Typo is in conclusion. 'EF up to 3.6×10^7 ' should be changed into ' 3.6×10^7 '.

Reviewer #1

Authors for the first time demonstrated that one of the carbon allotrope graphdiyne (GDY) shows remarkably Raman signal enhancement activity compared to graphene and other allotropes. The prepared GDY hierarchical hollow microspheres for this and tested R6G and showed the perspective for bisphenol A and dichlorophenol sensing. Also, they tried to explain the enhancement effect by interfacial charge transfer due to sp hybridized carbon with R6G. Despite the work is original and reporting first time Raman enhancement on GDY, there is no clarity about the existing/potential advantages compared to existed noble metal materials and recently reported 2d layered materials (MoS₂, Mxene, h-BN) etc.

Reply: We are very grateful to the reviewer for the positive comments on our work. Below, we reply to the reviewer's comments one by one.

1. Despite the preparation of a high surface area GDY is highly promising in general (catalysis, adsorbent etc), this substrate is not a perfect model for the evaluation of mechanistic aspects of Raman enhancement. This is because for there is a strong dependence of Raman signal intensity on the 2d materials (graphene and other) thickness [J. Am. Chem. Soc. 2018, 140, 8696–8704]. Taking into account the morphology of GDY hierarchical hollow microspheres, this is very difficult to evaluate thickness. This is not clear what was the thickness of SERS substrates with 0.8 cm² and if this thickness is homogeneous over the substrate. Unfortunately, there is no information about the preparation of SERS substrates for EF measurements. The preparation of GDY film has been reported according to the literature [Acc. Chem. Res. 2017, 50, 10, 2470–2478]. So, I suggest confirming these findings on GDY film or providing more detailed characterization of SERS substrate from microspheres.

Reply: We are very grateful to the reviewer for this comment, as well as for the two articles recommended by the reviewer (we have carefully read them and they have been added to the revised version). In the revision, we described in detail the preparation process of SERS substrate for EF calculation and the experimental parameters such as the mass, thickness and area of the substrate. We also provide a schematic diagram to facilitate readers to understand the preparation process. **For details, please see the experimental part of the Supporting Information (1.1. Calculation of Raman Enhancement Factor) and Figure S28.**

Figure S28. SERS substrate preparation diagram and prepared substrate photos for EF calculation.

Page 2 in Supporting Information: As shown in Figure S28, in the experiments, disperse 10 mg of GDY HHMSs in 3 mL of anhydrous ethanol and ultrasound for 2 min. Add the obtained uniform GDY suspension droplets into a glass circular groove with an area of about 0.8 m². After the ethanol volatilizes, drip the remaining suspension until all the suspension is used up. Finally, a layer of GDY substrate was formed in the groove, with a thickness of about 0.1 mm.

In addition, we would like to discuss these questions. First of all, as for the thickness of SERS substrate, it is generally believed that in the 2D semiconductor SERS substrate where chemical enhancement plays a dominant role, this thickness refers to the thickness of the 2D structure itself, such as the thickness of the nanosheets, rather than the thickness of the macro substrate. This is because the chemical enhancement mainly depends on the interface charge transport (ICT), while the ICT effect depends heavily on the thickness of the nanosheet. This rule has also been described in the literature [J. Am. Chem. Soc. 2018, 140, 8696–8704]. For example, the SERS properties of single-layer, double-layer and 5-layer MoTe₂ nanosheets are different. For the current GDY hollow spheres, the thickness of the nanosheets is about 2.5 nm (about 7 layers GDY), and this extremely thin thickness is favorable for ICT effect. At present, we are unable to control the thickness of the GDY nanosheets. We hope to achieve accurate control of the thickness of the GDY nanosheets in the future and measure the correlation between SERS properties and thickness. Secondly, it is a difficult problem to prepare smooth GDY nanofilms. The GDY films listed in the review literature [Acc. Chem. Res. 2017, 50, 10, 2470–2478] are actually hierarchical structures assembled from GDY nanosheets. We hope to prepare smooth GDY films in the future and observe their SERS properties. Once again, we thank the reviewer for the comments.

2. Authors are using R6G and Raman excitation of 532 nm, leading to the surface-enhanced resonant Raman scattering (SERRS) phenomenon, which occurs as excitation laser frequency crosses the frequencies of electronic excited states of the molecule. Although I consider this system not perfectly relevant for mechanistic study because a big part of signal enhancement is coming from this phenomenon, there are extensive examples of this situation. This fact should be underlined in the mechanistic study and SERS and SERRS should not be mixed. The positive part is that authors also showed the SERS activity of GDY for bisphenol A and dichlorophenol.

Reply: We are very grateful to the reviewer for this constructive suggestion, and we agree with the reviewer's views on SERRS. Because the energy gap (2.3 eV) of the R6G and the excitation energy (532 nm, 2.33 eV) are very close, SERRS will be generated. In order to eliminate the resonance enhancement between R6G and the excitation (532 nm), we added the SERS effect under 633 nm excitation in the revision. Because the energy (1.95 eV) of the 633 nm excitation is quite different from the energy gap (2.3 eV) of R6G, SERRS is effectively avoided. The experimental results showed that the Raman enhancement factor (EF) obtained under 633 nm excitation is 2.7×10^6 . We have added these discussions to the revised version.

Page 14, paragraph 1: Considering that the 532 nm excitation (2.33 eV) is very close to the energy gap of R6G (2.3 eV), in order to eliminate the contribution of molecular

resonance, we replaced the 532 nm excitation with 633 nm excitation (1.95 eV). The results showed that under 633 nm excitation, R6G molecules obtained EF of 2.7×10^6 on these GDY HHMSs.

3. The experimental details about SERS measurement in Method section and SI are confusing “30 mg of GDY HHMSs was dispersed into a probe solution (30 mL)” and “the dye solution with 10 mL volume was drop-casted on the SERS material with $\sim 0.8 \text{ cm}^2$ substrate area followed by a gentle dry process”. Please, explain, which procedure was used for EF measurement and SERS sensing of other molecules and why these procedures are different

Reply: We are very grateful to the reviewer for this comment. The SERS detection method mentioned in the Method Section of the main text is determined according to the actual detection requirements, because this method can maximize the use of the high adsorption of SERS materials with high specific area, thus obtaining the lowest detection limit (LDL). At present, SERS technology is mainly used for qualitative screening of hazardous substances in actual detection, so it is particularly important to obtain an excellent LDL, and our processing method can make GDY hollow spheres fully absorb analyte molecules as much as possible, so as to obtain the best LDL. Chinese Academy of Inspection and Quarantine is a practical application-oriented scientific research institution, so we have adopted this SERS detection method. As for the experimental method of calculating Raman EF in Supporting Information, we have adopted the method reported in the literature [J. Am. Chem. Soc. 2018, 140, 8696–8704] on this subject, so as to keep the consistency of the method as much as possible and facilitate the comparison of EF.

4. What are the general advantages of GDY for SERS compared to common metal-based plasmonic materials providing even higher EF factors?

Reply: We are very grateful for this good question put forward by the reviewer. In fact, this question is also a motivation to consider when we start the research of a new type of SERS active material. As the reviewer said, the Raman EF of some noble metal SERS substrates is very high, even reaching 10^9 level. However, noble metal as SERS substrate also has some inherent disadvantages, such as the high price of gold substrate, the easy oxidation of silver substrate, and the difficulty of obtaining high signal uniformity of noble metal nanostructure substrate. For GDY, as a carbon material, it has the advantage of natural raw materials. As the synthesis method becomes easier, its price will become lower and lower, which is convenient for large-scale application. In addition, GDY has high chemical stability and can be stored in air for a long time. In addition, due to its inherent layered structure, it is easy to obtain uniform signal uniformity on its surface. Finally, as a semiconductor SERS substrate, GDY can obtain high chemical selectivity by adjusting its band gap, which is very beneficial for selective detection. To sum up, GDY is a promising SERS substrate material. We have added these discussions to the revised version.

Page 19, paragraph 1: Considering the abundance of GDY raw materials, high chemical stability and convenient adjustment of energy band, GDY is very likely to become an excellent nonmetallic SERS substrate.

5. In case of drop deposition of R6G, the formation of the coffee ring effect is common [ChemPhysChem. 2015 16(13):2726-34], how authors addressed this point during EF measurement and comparison with graphene?

Reply: We are very grateful to the reviewer for this comment. The coffee ring effect is indeed a phenomenon that occurs easily when noble metal nanoparticles are dried, and we have also found it in previous experiments. In order to solve this troublesome problem, we specially designed a round groove with glass texture to prepare SERS substrate. Due to the space limitation of the groove, these SERS substrates can maintain their original appearance and avoid the occurrence of coffee ring phenomenon. Please See Figure S28 for details. In addition, in order to eliminate the influence of substrate heterogeneity as much as possible, we have taken the average value of signals at 10 positions when calculating EF. We have added the optical photos of GDY substrate to the Supporting Information.

Minor questions:

6. How authors showed Raman mapping in 2 cm^2 if the size of the sample was $\sim 0.8 \text{ cm}^2$? Please, provide this data as a coloured map for the sake of clarity.

Reply: We are very grateful for this good question put forward by the reviewer. In fact, 0.8 cm^2 is the area for Raman EF calculation, and 2 cm^2 is the area for measuring the relative standard deviation (RSD) of signals. The reason why a larger area is used to calculate RSD is to obtain more accurate signal uniformity. According to the suggestion of the reviewer, we have added the SERS mapping images to the Supporting Information.

Figure S22. SERS mapping images of the GDY/R6G. (a) R₁ at 612 cm^{-1} . (b) R₂ at 772

cm⁻¹.

7. Please, clarify more measurement details, the power 0.7 mW was used with laser spot 5 mm? This is not clear objective × 50 L could give so large spot?

Reply: For the beam spot diameter of 5 mm, the first letter “m” should actually be its corresponding Greek letter “μ”. We are very sorry for this writing error. The correct beam spot diameter is 5 μm. We have corrected this error in the revision.

8. Fig. 1b show Raman spectra of GDY, there are 2 peaks at 1932 and 2173 cm⁻¹. Could, you, please, clarify the origin of peak at 1932 cm⁻¹, this is not typical for C≡C.

Reply: This is a very good question. According to the suggestion of the reviewer, we consulted many reports on GDY characterization (*J. Am. Chem. Soc.* **2015**, *137*, 7596-7599; *Angew. Chem. Int. Ed.* **2022**, *61*, e202210242; *Angew. Chem. Int. Ed.* 2019, *58*, 746-750). The results shown that in the Raman spectra of GDY, these articles refer to these two peaks as acetylenic bond, therefore, we continue this statement. We have added the relevant references to the revision (***please see Referenes 32 and 37***).

For example:

[Redacted]

Figure 2e in Reference 32 (*J. Am. Chem. Soc.* **2015**, *137*, 7596-7599): Synthesis of Graphdiyne Nanowalls using Acetylenic Coupling Reaction.

9. Please, provide details of graphene origin used in this study.

Reply: We thank the reviewer for this suggestion. These graphene nanosheets for experiments were purchased from XFNANO company, which is a kind of reduced graphene oxide. This information has been added to the Supporting Information (***please see Figure S26***).

Reviewer #2

This article demonstrates spherical porous graphdiyne (GDY) synthesis by using and interface-induced growth mechanism and SERS application with rhodamine 6G (R6G). The spherical porous graphdiyne is a unique structure and homogeneous and it shows high SERS signals by charge transfer. The result looks interesting, but some should be clear for the publication with major revisions in Nature Communications.

Reply: We are very grateful to the reviewer for the positive comments on our work. Below, we reply to the reviewer's comments one by one.

1. Page 4, line 8: It is not only noble-metal substrates. Non-noble metals such as aluminum have also been investigated. And, other metals can be candidates for SERS substrates.

Please, check some refs. and add it to the manuscript.

1. Chem. Commun., 2014, 50, 3744-3746

2. Appl. Opt. 1995, 34, 4755-4767.

Reply: We are very grateful to the reviewer for this suggestion. These references have been added to the revision (***Please see reference 11 and 12***).

2. Page 4, line 9: Recently, Cu was also reported as SERS substrates.

And various structures should be mentioned because they are also an important factor for SERS enhancement.

Some structural effects of SERS may be very helpful for readers.

please check and add some refs.

1. Small 2022, 18, 2107182

2. The Journal of Physical Chemistry C 124 (43), 23730-23737

3. Journal of Materials Chemistry A 8 (40), 21016-21025

4. RSC advances 10 (14), 8309-8313

Reply: We are very grateful to the reviewer for this suggestion. These references have been added to the revision (***Please see reference 13-16***).

3. Page 4, line 11: SERS is not an electromagnetic phenomenon.

It originates from electric field resonance by coherent electron gas. Please double-check and correct it if needed.

Reply: We are very grateful to the reviewer for this suggestion. According to the suggestion, we have revised this sentence:

Noble-metal substrates, such as Au and Ag, can take advantage of the Localized field induced surface plasmon resonance (SPR), especially the emergence of a large number of "hot spots" (high-intensity electromagnetic field regions formed at nanoscale gaps), resulting in dramatic enhancement of SERS signals.

If there are still problems, we can further modify it.

4. Page 6, line 6 and 8 (Figure S1a, Figure S1b): No letter "a" and "b" in the Figure S1

Reply: We are very sorry for our carelessness. "a" and "b" have been marked in Figure S1.

Figure S1. Photos of two-phase liquid before (a) and after (b) reaction. It can be clearly seen that a large number of GDY products are formed on the two-phase interface.

5. Page 7, the last two lines: Could you please mention how you corrected the XPS peaks? In general, people use C1s orbital as a ref. But for carbon materials, a different ref. peak should be used to operate deconvolution of C1s.

Reply: This is a very good question, which is very important for XPS measurement of GDY. As the reviewer said, in general, people use C1s orbital (284.8 eV) as a ref, which is applicable to saturated carbon. The carbon that makes up GDY is unsaturated carbon, both sp and sp^2 , of which sp^2 hybrid carbon accounts for the majority. Therefore, different from the calibration method of saturated carbon, we use sp^2 hybrid carbon as the ref, which binding energy is usually 284.1 eV. We have explained the XPS measurement method in the revised version (***please see Material Characterization in Method***). Once again, we thank the reviewer for this constructive suggestion.

6. Page 8, line 1: Did you use a Cu TEM grid without any carbon layer for the TEM EDS?? You should mention which TEM grid you used. If you use any holey carbon grid, Figures S4 and 5 shouldn't be the same.

Reply: We are very grateful to the reviewer for reminding us. When we used TEM-EDS to measure the composition of the GDY hollow spheres, we mistakenly used the most common Cu grid, so there was a very strong Cu signal in the EDS spectrum (Figure S5), while there was no copper signal in the XPS spectrum (Figure S4). Other reviewers also pointed out this problem. According to the reviewer's comments, we have re-measured the EDS spectra of these GDY hollow spheres with carbon-free Mo grid, and it can be seen that they are copper-free. These new results have replaced the original figure (Figure S5).

Figure S5. EDS spectrum of the GDY HHMSs. Molybdenum signal comes from molybdenum grid.

6. Page 8, line3-5: Please conduct O 1s deconvolution as well.

Reply: We are very grateful to the reviewers for their suggestions. The O 1s spectrum has been added to the support information, please see Figure S6.

Figure S6. O 1s spectrum of the GDY HHMSs.

8. Page 8, line12: Scanning electron microscopy (SEM) images shows  show

Reply: We are very grateful to the reviewer for carefully pointing out our grammatical error, which has been corrected.

9. Page 8, line14: It is hard to say that (2.5 nm thickness). Because the resolution of SEM is generally more than 2.5 nm, although the spec sheet says it is below 2.5 nm. Please

check again.

Reply: We are very grateful to the reviewers for this suggestion and agree with his (her) view. At present, we have used SEM and AFM to characterize the thickness of these GDY nanosheets. Unlike SEM, AFM can more accurately measure the thickness of nanosheets. The AFM measurement results (Figure S8) are basically consistent with the SEM measurement results (Figure S8). According to the reviewer's suggestion, in the revision, we also used TEM to measure the thickness of these nanosheets (Figure S11), and the results are also highly consistent with AFM. Therefore, the thickness of these GDY nanosheets is relatively accurate. Once again, we thank the reviewers for their suggestions.

Figure S11 (c) Thickness measurement by the cross-section of the nanosheets.

10. Page 11, Chapter "SERS Properties of GDY HHMSs"

How did you count the number of R6G on the SERS substrates? The EF can be easily overestimated due to the underestimated number of molecules. Especially, it is even easier to be underestimated when a material with a high surface is used.

Please describe the details of how you prepared the sample and the reference for the SERS.

Reply: We are very grateful to the reviewer for this comment. In fact, the accurate calculation of Raman EF has always been a difficult problem. Internationally, scientists engaged in SERS research have also been calling for the adoption of a unified standardized method to calculate EF, but until now, the international standard method for EF calculation has not been promulgated. At present, we all use the general methods in the literature. As for EF computing, we also discussed with Professor Bin Ren of Xiamen University. As an excellent SERS scientist, he also hopes to publish an international standard for EF computing in the future. However, it is really difficult to achieve this goal. For example, the statistics of molecular number adsorbed on SERS substrate is a very difficult problem, especially considering the ultrahigh adsorption of materials with high specific surface area. Therefore, when we calculate the EF of these GDY hollow spheres, we also use the method commonly used in the current literature. We hope to introduce a

standard method to solve this problem in the near future.

11. Page 18-20, Chapter "Method"

There are many typos. Please check if you have done superscripts or subscripts. An entire English check should be done.

Reply: We are very grateful to the reviewer for carefully pointing out our grammatical error and typos. According to the suggestion of the reviewer, we carefully checked the grammar and writing of the full text and corrected the errors found.

Reviewer #3

This manuscript reports the graphdiyne (GDY) hierarchical hollow microsphere (HHMS) as a highly sensitive surface-enhanced Raman scattering (SERS) substrate by virtue of its strong chemical enhancement and high specific surface area. The synthesis process is interesting while the quality of the synthesized HHMS is high. The substrate shows excellent SERS performance for several analytes, including R6G, BPA, and 2,4-DCP. Moreover, the authors tried to elucidate its enhancement mechanism by proposing the charge-transfer diagram based on several experimental evidences and density function theory (DFT) calculation. The results are interesting and promising. However, I have several major concerns (see below). I will consider recommending its publication in Nature Communications if the authors can address the following concerns convincingly.

Reply: We are very grateful to the reviewer for the positive comments on our work. Below, we reply to the reviewer's comments one by one.

1. The authors claim that their GDY may be the first truly all-carbon semiconducting SERS substrate, which is incorrect. Diamond has been reported as an all-carbon semiconducting SERS substrate [Gao, Ying, et al. "Semiconductor SERS of diamond." *Nanoscale* 10.33 (2018): 15788-15792.]. In addition, although perfect large-area graphene is not a semiconductor while its nanostructures are semiconductors and have been reported for SERS [Das, Ruma, et al. "Origin of high photoluminescence yield and high SERS sensitivity of nitrogen-doped graphene quantum dots." *Carbon* 160 (2020): 273-286.].

Reply: We are very grateful to the reviewer for pointing out the inaccuracy of our statement. We have revised the original statement and deleted this statement in the revised version. And we also added these references to the revision, ***please see References 22 and 23***. Once again, we thank the reviewer for the comment.

2. From the EDS spectrum in Figure S5, it is evident that Cu which is a well-known SERS active element exists in the synthesized GDY HHMS. The authors should exclude its contribution to the observed SERS performance.

Reply: We are very grateful to the reviewer for reminding us. When we used TEM-EDS to measure the composition of the GDY hollow spheres, we mistakenly used the most common copper grid (without carbon), so there was a strong copper signal in the EDS spectrum (Figure S5), while there was no copper signal in the XPS spectrum (Figure S4). Other reviewers also pointed out this problem. According to the reviewer's comments, we have re-measured the EDS spectra of these GDY hollow spheres with carbon-free

molybdenum grid, and it can be seen that they are copper-free. These new results have replaced the original figure (Figure S5).

Figure S5. EDS spectrum of the GDY HHMSs. Molybdenum signal comes from molybdenum grid.

3. In addition to R6G, the authors claim that their substrate also shows excellent SERS performance for BPA and 2,4-DCP. What are the enhancement factors for these two molecules? Is the proposed enhancement mechanism for R6G also applicable for these two molecules while the HOMOs and LUMOs of these two molecules are quite different from that of R6G?

Reply: We are very grateful to the reviewers for their comments, which is a very good suggestion. According to this suggestion, we estimated the Raman EF of these two molecules on the GDY hollow sphere, which is 100 and 90 respectively. At present, we temporarily believe that the Raman enhancement of these two molecules is also related to the interface charge transfer. DFT calculation reveals that GDY possesses rich energy levels from the -2 – 2 eV region (Figure 5a). The LUMO level and HOMO level of BPA are 0.15 and -0.31 eV, respectively. Therefore, the energy required for molecular transition is only 0.46 eV for BPA. Since the excitation wavelength is 532 nm (2.32 eV), it is sufficient to drive charge transition from HOMO to LUMO. Furthermore, the rich energy level (-2 – 2 eV) distribution of the GDY provides as many charge transfer channels as possible for the BPA/GDY system. Considering that these conclusions have not been fully confirmed, we have not added them to the revision. Our current work mainly focus on the synthesis of a novel GDY nanostructure and the discovery of their SERS properties. We hope to find out its enhancement mechanism for different molecules in the next in-depth study. Once again, we thank the reviewers for the comment.

4. In Figure 4c, the authors claim that the newly formed absorption peak at around 461

nm indicates the charge-transfer between GDY and R6G. However, compared with the UV-Vis spectrum of GDY HHMS in Figure S7, it seems that this peak is just the peak of GDY HHMS rather than a newly formed peak.

Reply: We are very grateful to the reviewers for this comment. Indeed, as the reviewer said, the UV-Vis absorption spectrum of GDY (Figure S7) is similar to that of GDY-R6G (Figure 4c). However, by carefully comparing the two figures, it can be found that the strongest absorption position of GDY-R6G (461 nm) has a significant blue shift compared with GDY (507 nm). Once again, we thank the reviewer for this good comment.

Figure S6

Figure 4c

5. The explanation of the enhancement mechanism is not clear. Based on Figure 5d, the authors claim that the main charge accumulation and depletion only happened inside of R6G when R6G is adsorbed on graphene. How did the authors get this conclusion? Does this conclusion contradict with the excellent fluorescence quenching capability of graphene observed in experiment and in other references.

Reply: Thanks for reviewer's professional comment. As discussed above, SERS originates from electric field resonance by coherent electron gas, in this work, the coherent electron gas is mainly contributed by the ICT effect. Figure 5d clearly reveal that the charge accumulation region (marked as yellow) and depletion region (marked as blue) both mainly locate at R6G molecule and there is a little charge accumulation region locating at graphene, conversely, Figure 5c clearly shows that charge accumulation region (marked as yellow) remarkably cover on GDY and R6G. Accordingly, we revise the conclusion to “the main charge accumulation and depletion mainly happened inside of R6G when R6G is adsorbed on graphene” in manuscript.

This conclusion is not contradicted with the excellent fluorescence quenching capability of graphene. Although the ICT effect of R6G on graphene is much weaker than that on GDY, however, there is still 0.72e transferring from R6G to graphene. As described in [Xie, Liming, et al. Graphene as a substrate to suppress fluorescence in resonance Raman spectroscopy. *J. Am. Chem. So.* 2009, 131, 9890-9891], the excellent fluorescence quenching capability of graphene is due to the intrinsic metallic of graphene with delocalized π electrons, the excited state of R6G molecule can still transfer electron to graphene and result in suppressing fluorescence. Consequently, the ICT effect of R6G

on graphene can suppress fluorescence but form weak electric field resonance between R6G and graphene. And we further revised manuscript accordingly (**please see page 16-18**). Once again, we thank the reviewer for the comment to make this point clearly.

6. Based on the DFT calculation, the authors claim that the magnetic moment of R6G disappears when it is adsorbed on GDY. What occurs behind this should be clarified. How the magnetic moment of R6G influences the Raman enhancement should be discussed in detail as well.

Reply: Thanks reviewer's kind comment. According our calculation, the isolated R6G molecule possesses a magnetic moment of 1 μ_B /molecule (similar with O_2 molecule), means that there exists an unpaired electron in molecule. And the magnetic moment of R6G disappears when it is adsorbed on GDY, meanwhile, there is 1.06e transferring from R6G to GDY, conversely, the magnetic moment of R6G does not disappear when it is adsorbed on graphene and resulting weak electric field resonance. Hence, we can conclude that this unpaired electron transfers to GDY and results in remarkable ICT effect, inducing strong electric field resonance, herein we can conclude that magnetic moment of R6G is crucial for SERS in this work. Accordingly, we further revised manuscript (**please see page 16-18**).

7. The authors argue that the sp and sp^2 hybridized carbon atoms are crucial for SERS in their manuscript. A more detailed discussion should be given to justify this.

Reply: Thanks for reviewer's professional comment. Notably, the electrons in sp hybridized carbon atoms is more local than that in sp^2 hybridized carbon atoms because $C\equiv C$ is stronger than $C=C$ and the bond length of $C\equiv C$ is shorter than $C=C$, hence, there is more peaks in PDOS of GDY near Fermi level than that of graphene (Figure 5a and Figure 5b), the totally delocalized sp^2 hybridized carbon atoms in graphene is not benefitted to match the energy level for R6G. The sp hybridized carbon atoms in GDY provide more energy level to match the energy level of R6G and form remarkable electric field resonance, resulting stronger SERS than that of graphene. Accordingly, we further revised manuscript (**please see page 16-18**).

8. The caption of Figure 5 does not match with the figure panels.

Reply: We are very grateful to the reviewer for pointing out our error, and we apologize for our carelessness. This error has been corrected in the revision.

Reviewer #4

The authors demonstrated a liquid-liquid interface-induced method to synthesize graphdiyne porous flower like particles and showed SERS capabilities to detect R6G at 10^{-11} M level. The major concern of the work is that Cu contamination can be noticed via the provided EDX and XPS. Cu nanoparticles also display SERS; thus, therefore it is not clear that only graphdiyne is responsible for the SERS detection limit. Thus, I cannot recommended this paper for publication in Nature Communications.

Reply: We are very grateful to the reviewer for the constructive comments on our work. It was just under this comment that we suddenly realized that we mistakenly used copper

grid as the sample support film in the TEM-EDS measurement, so there was a strong copper signal. We have replaced the copper grid with molybdenum grid and conducted EDS measurement again. The result shows that the sample does not contain copper. Below, we reply to the reviewer's comments one by one.

1. In Figure 1b, the authors have showed the Raman spectra of the as synthesized graphdiyne, other than the D ($\sim 1397\text{ cm}^{-1}$), G ($\sim 1577\text{ cm}^{-1}$), and Y-band ($\sim 2173\text{ cm}^{-1}$), there is a band with relatively high intensity around 1932 cm^{-1} . However, it is noticed that in some literature, the peak around 1932 cm^{-1} is missing (Appl. Phys. Lett. 113, 021901 2018; DOI: 10.1126/sciadv.aat6378). Could the authors explain the peak around 1932 cm^{-1} ?

Reply: This is a very good question. According to the suggestion of the reviewer, we consulted several reports on GDY characterization (such as *J. Am. Chem. Soc.* **2015**, *137*, 7596-7599; *Angew. Chem. Int. Ed.* **2022**, *61*, e202210242; *Angew. Chem. Int. Ed.* 2019, *58*, 746-750). The results shown that in the Raman spectra of GDY, these articles refer to these two peaks as acetylenic bond, therefore, we continue this statement. We have added the relevant references to the revision. As for why the peak at 1932 cm^{-1} disappeared in some references, we haven't thought of a possible answer yet. But this is really a problem worthy of in-depth study. We thank the reviewer for providing us with new research direction. We have added the relevant references to the revision (**please see Referenes 32 and 37**).

For example:

Figure 2e in Reference 32 (*J. Am. Chem. Soc.* **2015**, *137*, 7596-7599): Synthesis of Graphdiyne Nanowalls using Acetylenic Coupling Reaction.

2. Considering the possible metal or metal oxide contamination, Could the authors also show the Raman spectra starting from 500 cm^{-1} ?

Reply: We are very grateful to the reviewer for this comment, which is a constructive suggestion. According to the suggestion, we expanded the range of Raman spectrum, and the results showed that there was no copper oxide in these GDY hollow spheres. The new Raman spectra have been added to the revision (**please see Figure 1b**).

3. XPS shown in Figure S4, peaks above 950 eV can be found. Could that be the Cu contamination from the synthesis?

Reply: We are very grateful to the reviewers for pointing out our omissions. Because the synthesis of GDY requires Cu^{2+} , these Cu^{2+} is easily hydrolyzed into CuO in aqueous solution (this phenomenon has also been reported by other researchers, see “A Deprotection-free Method for High-yield Synthesis of Graphdiyne Powder with In Situ Formed CuO Nanoparticles. *Angew. Chem. Int. Ed.* **2022**, *61*, e202210242”). Therefore, we use hydrochloric acid to dissolve these possible copper oxide when cleaning the sample. After careful examination of the experimental records and samples, we unexpectedly found that the samples used for XPS detection were not washed with hydrochloric acid, so they contained a part of copper oxide nanoparticles hydrolyzed from Cu^{2+} ions. We re-prepared the sample and ensured that there was no CuO in it through dissolution of hydrochloric acid. In the revision, the original XPS spectrum has been replaced by the corrected XPS spectrum. It should be pointed out that the samples used for SERS test does not contain CuO. In addition, it should be noted that CuO usually has no SERS effect.

Figure S4. XPS survey spectrum of the GDY HHMSs.

In the TEM image of the sample before cleaning with hydrochloric acid, many CuO

nanoparticles can be seen attached to the GDY nanosheets.

4. The authors used EDX (Figure S5) to show/confirm the C/O ratio. However, EDX is not the ideal characterization technique to quantify light elements like C and O. It would be better to use EELS or XPS, instead of EDX. EDX is good for heavy element, such as Cu. It is clear that Cu is found within the samples based on Figure S5.

Reply: We are very grateful to the reviewer for reminding us. When we used TEM-EDS to measure the composition of the GDY hollow spheres, we mistakenly used the most common copper grid (without carbon), so there was a strong copper signal in the EDS spectrum (Figure S5), while there was no copper signal in the XPS spectrum (Figure S4). According to the reviewer's comments, we have re-measured the EDS spectra of these GDY hollow spheres with carbon-free molybdenum grid, and it can be seen that they are copper-free. These new results have replaced the original figure (Figure S5).

Figure S5. EDS spectrum of the GDY HHMSs. Molybdenum signal comes from molybdenum grid.

5. The authors mentioned that graphdiyne is a direct band gap semiconductor, thus, it is recommended to show the photoluminescence, other than UV-Vis.

Reply: We are very grateful to the reviewers for their comments. At the beginning of this work, we also wanted to measure the photoluminescence (PL) spectra of these GDY hollow spheres, but unfortunately, we did not get them. We have looked up a large number of research papers on GDY and found no PL on GDY nanostructures. In almost all the references, the band gap of GDY is obtained by UV-Vis spectrum. At present, we do not know the reason for this phenomenon, but we are very grateful to the reviewers for putting forward a direction worthy of in-depth study.

6. The morphology of the graphdiyne is the porous flower like particles. How did the authors measure the flake thickness of the particles?

Reply: We are very grateful to the reviewer for this important question. We used three methods, SEM (Figure S8), AFM (Figure S9), and Figure S11 to measure the thickness of these GDY nanosheets. For SEM and TEM, we only need to measure the thickness of the cross section of these nanosheets in the high-magnification images. For AFM measurement, we first let some nanosheets fall off the hollow spheres through strong ultrasound, and then measure the thickness of these individual nanosheets.

7. In the characterization section, many details are missing. For example, the excitation wavelength of the Raman measurements, the accelerating voltage for SEM and HRTEM, etc.

Reply: We are very grateful to the reviewer for pointing out the lack of experimental information. We have added detailed experimental information in the revised version.

Reviewer #5

The article “Liquid-liquid interface growth of graphdiyne with ultrahigh specific surface area as highly sensitive surface-enhanced Raman scattering substrate” proposed a new micro carbon particle made with graphdiyne, which is composed of hollow core with buddied nanosheets. They emphasized that they conducted the SERS study for the first time as a microparticle made of GDY, and obtained about 1000 times stronger SERS signal than the existing carbon-based particles. In addition, it was theoretically revealed that this SERS enhancement was caused by the ICT mechanism. This is an interesting study and a meaning addition to the literature in related fields with some novel aspects. There are, however, some serious concerns and issues that need to be fully addressed and the manuscript needs to be significantly improved to be considered for publication in Nature Communications. In the current form, this paper fits in a bit more specialized journal.

Reply: We are very grateful to the reviewer for the positive comments on our work. Below, we reply to the reviewer's comments one by one.

1. The author stated that Cu^{2+} ion was used as a catalyst, but the EDS result of the product showed a high Cu content (Figure S5) with more than half of carbon in atomic ratio. This seems to prove that this particle is not a pure carbon particle as claimed by the authors, but a composite particle containing an excessive amount of Cu. Also, the color of the product is brown, further showing the existence of copper.

Reply: We are very grateful to the reviewer for the constructive comments on our work. It was just under this comment that we suddenly realized that we mistakenly used copper grid as the sample support film in the TEM-EDS measurement. When we used TEM-EDS to measure the composition of the GDY hollow spheres, we mistakenly used the most common Cu grid, so there was a strong copper signal in the EDS spectrum (Figure S5), while there was no copper signal in the XPS spectrum (Figure S4). According to the reviewer's comments, we have re-measured the EDS spectra of these GDY hollow spheres with carbon-free Mo grid, and it can be seen that they are copper-free. These new results have replaced the original figure (Figure S5).

Figure S5. EDS spectrum of the GDY HHMSs. Molybdenum signal comes from molybdenum grid.

As for the color of the sample, the GDY product shown in Figure S1 has not been washed and contains a large amount of Cu^{2+} . In addition, the by-products and the refraction of liquid and glass will also affect the color. Under the joint influence of these factors, the product shows dark brown. In fact, after washing, the product is black. We have added the photo of the washed GDY hollow spheres to Figure S1.

We fully understand the reviewer's important attention to this issue, and we apologize for our carelessness.

Figure S1. Photos of two-phase liquid before (a) and after (b) reaction. It can be clearly seen that a large number of GDY products are formed on the two-phase interface. (c) The photo of the GDY sample after cleaning.

2. The quantitative signal of this particle was claimed by comparing the commercial AuNPs and the SERS signal, but it is not that meaningful nor fair because it was compared with the worst case (Figure S20a).

Reply: We very thank the reviewers for their comments, and we also agree with the reviewer's view. If necessary, we are willing to delete Figure S20a.

4. At the EF calculation equation in the supporting information, if I'm not mistaken, the calculation with the numbers and equations given by the authors do not reproduce the given EF of 3.6×10^7 , but suggests EF of 4.4×10^8 . Please recheck the number.

Reply: We are very grateful to the reviewer for pointing out our mistakes. After careful examination of the calculation process, we found that the cause of this error was the wrong R6G data. The molecular weight of R6G is 479.01 instead of 442.5, and its density is 1.15 g cm^{-3} instead of 1.26 g cm^{-3} . We may filled the molecular weight and density of another molecule under study into the equation. In addition, the concentration of R6G we used was $1 \times 10^{-10} \text{ M}$, and we mistakenly filled in $1 \times 10^{-11} \text{ M}$. We have corrected these mistakes in the revision. We are very sorry for our carelessness.

5. The notes say that the peak intensity was acquired with 15 s acquisition time in GDY, but when the Raman intensity ratio was estimated, an equation of $(31.5/40)/(421.5/0.5)$ was used, suggesting that 40s acquisition time was used in the calculation.

Reply: We are very grateful to the reviewer for pointing out our mistakes. 40 s is correct. We are very sorry for our carelessness.

6. In the EF calculation equation in the supporting information, r and ρ are both used as a notation for the density of bulk dye. It might be beneficial to unify these

Reply: We are very sorry for our carelessness. We forgot to convert r into its Greek alphabet ρ . We have corrected this error in the revision.

7. The authors used a laser penetration depth of $21 \mu\text{m}$, possibly brought from the reference. However, the laser penetration depth is different from facility to facility, as they depend on the pinhole size and the objective lens used. Reliable reasoning and experimental values to confirm the laser penetration depth should be provided.

Reply: We are very grateful to the reviewer for this important suggestion. It is very difficult to measure the penetration depth of the laser, which involves complex issues such as the material type of the substrate, the size and morphology of the nanoparticles, the wavelength of the laser, and the power of the laser. So far, this problem has not been solved. Therefore, we quoted a reported value.

In fact, the accurate calculation of Raman EF has always been a difficult problem. Internationally, scientists engaged in SERS research have also been calling for the adoption of a unified standardized method to calculate EF, but until now, the international standard method for EF calculation has not been promulgated. At present, we all use the general methods in the literature. As for EF computing, we also discussed with Professor Bin Ren of Xiamen University. As an excellent SERS scientist, he also hopes to publish an international standard for EF computing in the future. However, it is really difficult to achieve this goal. For example, the statistics of molecular number adsorbed on SERS substrate is a very difficult problem, especially considering the ultrahigh adsorption of materials with high specific surface area. Therefore, when we calculate the EF of these

GDY hollow spheres, we also use the method commonly used in the current literature. We hope to introduce a standard method to solve this problem in the near future.

8. Adding the line slice image of AFM in Figure S8 would help readers to confirm the thickness of the nanosheets.

Reply: We are very grateful to the reviewer for the important suggestion. The line slice image have been added into Figure S9 in the revision.

Figure S9. AFM image of a single ultrathin nanosheet from the GDY HHMSs.

9. The authors projected RSD analysis in Figure 3c. In addition to that, giving a mapping image of SERS signal could also be a good way to show the uniformity of the substrate.

Reply: We are very grateful to the reviewer for the constructive suggestion. The SERS mapping image have been added into the Supporting Information.

Figure S22. SERS mapping images of the GDY/R6G. (a) R_1 at 612 cm^{-1} . (b) R_2 at 772 cm^{-1} .

10. A typo is in Figure S8. 'Sinlge' should be changed into 'single'.

Reply: We are very sorry for our carelessness. This typo has been corrected.

11. A Typo is in conclusion. 'EF up to 3.6×10^7 ' should be changed into ' 3.6×10^7 '.

Reply: We are very sorry for our carelessness. This typo has been corrected.

REVIEWER COMMENTS

Reviewer #1 (Remarks to the Author):

Authors addressed most of states issues, however, there are some critical questions. I believe that author should make some efforts to show that GDY is advantageous substrates compared to existed ones not only in terms of EF.

1. Please, add color scalebar to Figure S22 and add calculation details about RSD. Also, I still misdoubt about the high reproducibility (in terms of signal intensity) because of preparation way of SERS substrate (drop deposition of GDY to glass circular groove and uncontrolled evaporation). As author claim as advantage of GDY high reproducibility, sample to sample reproducibility should be added to discussion.

2. Authors claimed that potential advantages of GDY is “GDY has high chemical stability and can be stored in air for a long time”. However, experimental confirmation should be provided to support this assumptions

3. Regarding semiconductor nature of SERS substrate and possible high chemical selectivity by adjusting its band gap. I suggest to author provide some specific examples (relevant mixtire of molecules, where only one will be detected by GDY) of this principle and discuss it as an outlook.

Reviewer #2 (Remarks to the Author):

I believe the author(s) has answered and revised properly as the reviewers suggested. I recommend publication of this article if some are clearer.

1. The author tried to reveal the mechanism of the enhanced Raman signal. This is because there are other factors increasing the intensity such as underestimated molecular numbers. I understand this is not easy work but still want them make it clear to show the mechanism they assume is correct.

2. They used a SP2 orbital for XPS correction. SP2 is very close to C1s orbital, and not sure if

it is the right one. If I was them I would use slight Pt coating as a ref.

Reviewer #3 (Remarks to the Author):

This is the revised version of this manuscript. The authors addressed most of my comments, but their response to my two previous comments are still not convincing.

1. The authors' explanation of the enhancement mechanism for BPA and 2,4-DCP is not convincing as it is neither experiment-based nor theory-based. I suggest the authors at least show some theoretical evidence by DFT calculation or experimental evidence by absorption spectrum measurement to justify their claim of interfacial charge transfer.

2. In the response letter and the revised manuscript, the authors mentioned electric field resonance several times. What is this? Does this mean electromagnetic enhancement in SERS? Why does this so-called electric-field resonance occur?

Reviewer #5 (Remarks to the Author):

Overall, the authors addressed my major concerns well, and the paper has been improved significantly. This reviewer now recommends publication of this work in Nature Communications.

Reviewer #1

Authors addressed most of states issues, however, there are some critical questions. I believe that author should make some efforts to show that GDY is advantageous substrates compared to existed ones not only in terms of EF.

Reply: We are very grateful to the reviewer for the positive comments and concerns on our work. Next, we will reply to the reviewer's questions one by one.

1. Please, add color scalebar to Figure S22 and add calculation details about RSD. Also, I still misdoubt about the high reproducibility (in terms of signal intensity) because of preparation way of SERS substrate (drop deposition of GDY to glass circular groove and uncontrolled evaporation). As author claim as advantage of GDY high reproducibility, sample to sample reproducibility should be added to discussion.

Reply: We are very grateful to the reviewer for the suggestions. The color scalebar and RSD calculation method have been added to the revised version. With regard to this excellent reproducibility, it is reasonable to believe that it is closely related to the enhancement mechanism of semiconductor substrate and the two-dimensional (2D) structure of GDY. Specifically, the enhancement mechanism of the semiconductor substrate belongs to the interface-charge-transfer (ICT), so the signal intensity is closely related to the substrate/molecular interface structure. For GDY, its intrinsic 2D structure determines that its surface is very flat, which creates a very suitable condition for uniform ICT behaviour. This is completely different from the electromagnetic resonance enhancement mechanism of noble-metal substrates. For the noble-metal substrates, the signal intensity varies greatly due to the inhomogeneity of the electromagnetic field intensity caused by the inhomogeneity of the nanostructure. Only molecules falling on the "hot spot" of the electromagnetic field can obtain high strength Raman signals. According to the suggestion of the reviewer, we conducted the signal repeatability test on the other two substrates of the same batch, and the results showed that the RSD values obtained is similar to the RSD value obtained from the first substrate. Relevant discussions (page 13) and images (Figure S22) have been added to the text and Supporting Information.

Figure S23. SERS mapping images of the GDY/R6G. (a) R_1 at 612 cm^{-1} . (b) R_2 at 772 cm^{-1} .

Figure S22. RSD values of the SERS signal intensities obtained from the other two substrates of the same batch.

2. Authors claimed that potential advantages of GDY is “GDY has high chemical stability and can be stored in air for a long time”. However, experimental confirmation should be provided to support this assumptions

Reply: This is a very good suggestion. In order to prove the high chemical stability of these GDY samples, we investigated their oxidation resistance. The experimental results show that they have high chemical stability. Relevant experimental images have been added to the text and Supporting Information (Figure S30).

Figure S30. (a) XRD pattern and Raman spectrum (inset) and (b) SEM image of the GDY HHMSs after 3 months in air. These characterization results demonstrate that the structure and morphology of GDY HHMSs have not undergone significant changes, indicating their high chemical stability.

3. Regarding semiconductor nature of SERS substrate and possible high chemical selectivity by adjusting its band gap. I suggest to author provide some specific examples (relevant mixture of molecules, where only one will be detected by GDY) of this principle and discuss it as an outlook.

Reply: We thank the reviewer for the constructive suggestions. In the revision, we reviewed the typical researches on the high selectivity of semiconductor SERS substrates and looked to the prospect of GDY for selective detection.

Page 19 (yellow section): Moreover, the energy band structure of GDY can be adjusted by means of element doping and surface modification,^{29,30} which provides the possibility for selective molecular detection. This has been preliminarily proved on other semiconductor SERS substrates, such as metal-organic-framework compounds and oxygen-deficient nanostructures.^{7,46}

Reviewer #2

I believe the author(s) has answered and revised properly as the reviewers suggested. I recommend publication of this article if some are clearer.

Reply: We are very grateful to the reviewer for the positive comments on our improved version. Next, we will reply to the reviewer's concerns one by one.

1. The author tried to reveal the mechanism of the enhanced Raman signal. This is because there are other factors increasing the intensity such as underestimated molecular numbers. I understand this is not easy work but still want them make it clear to show the mechanism they assume is correct.

Reply: We thank the reviewer for the understanding of the difficulty of our work. Although we cannot directly and accurately calculate the number of adsorbed molecules at present, we have calculated the Raman EF of GDY/R6G molecules with lower concentration in this revision. The results show that the EF obtained at the concentration of 10 is basically the same as that at the concentration of 100, which further proves the accuracy of the GDY-based EF from the side. We also refer to the relevant literature (Raman EF calculation method), and our calculation process is

basically the same as that in the literature. Relevant discussions have been added to the revision. Once again, I would like to thank the reviewers for their constructive suggestions.

2. They used a SP2 orbital for XPS correction. SP2 is very close to C1s orbital, and not sure if it is the right one. If I was them I would use slight Pt coating as a ref.

Reply: We are very grateful for the correction method provided by the reviewer. We are learning this new correction method, which may take some time. We hope to learn this method and use it in our future research. Once again, I would like to thank the reviewer for his generous suggestion.

Reviewer #3

This is the revised version of this manuscript. The authors addressed most of my comments, but their response to my two previous comments are still not convincing.

Reply: We are very grateful to the reviewer for the positive comments on our improved version. Next, we will reply to the reviewer's concerns one by one.

1. The authors' explanation of the enhancement mechanism for BPA and 2,4-DCP is not convincing as it is neither experiment-based nor theory-based. I suggest the authors at least show some theoretical evidence by DFT calculation or experimental evidence by absorption spectrum measurement to justify their claim of interfacial charge transfer.

Reply: Thanks for reviewer's professional comment on the enhancement mechanism for BPA and 2,4-DCP. Accordingly, we further explore the charge interaction in GDY-molecules system via first principle calculations, the results clearly show that high-efficiency ICT within GDY-BPA and GDY-2,4-DCP system (Figure S29). The calculated results clearly reveal that the interaction electrons distributed on the GDY surface and BPA (2,4-DCP) molecule, and the ICT effect is significant in SERS mechanism of BPA and 2,4-DCP. Notably, we further demonstrate that the flat 2D structure of GDY creates a very suitable condition for uniform ICT behavior, is

completely different from the electromagnetic resonance enhancement mechanism of noble-metal substrates. Relevant discussions and pictures have been added to the text and supporting information.

Page 18 (yellow section): The interaction between GDY and BPA (2,4-DCP) is shown in Figure S29, the calculated results clearly reveal the same conclusion that the ICT effect play a curial role in SERS mechanism.

Figure S29. Charge density difference of (a) BPA and (b) 2,4-DCP adsorbing on GDY, where the yellow and cyan color mark the region of charge accumulation and depletion, respectively, the isosurface was set to $0.00015 \text{ e}/\text{\AA}^3$.

2. In the response letter and the revised manuscript, the authors mentioned electric field resonance several times. What is this? Does this mean electromagnetic enhancement in SERS? Why does this so-called electric-field resonance occur?

Reply: Thanks for reviewer's kind comment on this concept. After careful reconsideration, we do agree that concept of "electric field resonance" is ambiguous, the ICT effect, proposed in this work, is significant for SERS mechanism. Accordingly, we remove this concept in revised manuscript, and thanks for reviewer's careful correction to make the concepts more clearly to readers in this work.

Reviewer #5

Overall, the authors addressed my major concerns well, and the paper has been improved significantly. This reviewer now recommends publication of this work in Nature Communications.

Reply: We sincerely thank the reviewer for recommending the publication of our article.

REVIEWER COMMENTS

Reviewer #1 (Remarks to the Author):

Authors mostly revised my comments, however, I need to ask for some clarifications of their replies to improve the quality of the paper.

1. The question about chemical stability: please, please, show XRD and Raman spectra of freshly prepared graphdiyne on one graph for easier comparison. Please, add XPS of this samples, because C=O component and C/O ratio are more evident for confirmation of stability

2. About the chemical selectivity: both references (7, 46) are discussing sensing of dyes (R6G and MB), which are model analyte and not the most relevant molecules for real life application. So, I ask authors to suggest some specific molecules taking into account electronic structure of graphdiyne

Reviewer #2 (Remarks to the Author):

They answered my questions sincerely and made reasonable inferences, although the mechanism of the Raman enhancement effect is not yet 100% certain. Therefore, I think it would be acceptable for this paper to be published in Nature Communications.

Reviewer #3 (Remarks to the Author):

I am still concerned about the enhancement mechanism for BPA and 2,4-DCP. Their additional DFT calculation in this round of revision is too ambiguous and only shows the charge-density distributions between the analytes and GDY, which provides little evidence of ICT. Based on their DFT calculations, the authors should provide ICT pathways similar to those shown in Figure 4b and clarify their relations with the photon energy of the excited light. I cannot recommend the publication of this manuscript if the authors cannot provide a convincing explanation of the enhancement mechanism for BPA and 2,4-DCP.

Reviewer #5 (Remarks to the Author):

This revised paper can be published in Nature Communications without further revisions.

Reviewer #1

Authors mostly revised my comments, however, I need to ask for some clarifications of their replies to improve the quality of the paper.

Reply: We are very grateful for the positive comments from the reviewer on our revision, and we also appreciate the new questions raised by the reviewer to improve the quality of the article. Below, we will respond to the reviewer's questions one by one.

1. The question about chemical stability: please, show XRD and Raman spectra of freshly prepared graphdiyne on one graph for easier comparison. Please, add XPS of this samples, because C=O component and C/O ratio are more evident for confirmation of stability

Reply: This is a very good suggestion, and we have made modifications to the image layout based on the reviewer's suggestion.

Figure S31. (a) XRD patterns and (b) Raman spectra of the GDY HHMSs before and after 3 months in air.

At the same time, we have supplemented the XPS spectra of the GDY HHMSs after 3 months in air, which further demonstrate their high stability.

Figure S32. XPS spectra of the GDY HHMSs (a) after 3 months in air and (b) original.

2. About the chemical selectivity: both references (7, 46) are discussing sensing of dyes (R6G and MB), which are model analyte and not the most relevant molecules for real life application. So, I ask authors to suggest some specific molecules taking into account electronic structure of graphdiyne

Reply: Thank you very much for your interest in chemoselectivity. Indeed, R6G and

MB are commonly used model analytes, and we also recognize their limitations in practical applications. Therefore, we very much agree with your point of view that specific molecules match the electronic structure of GDY. In practical applications, we need to select suitable molecules for research based on specific needs and conditions. When considering the electronic structure of GDY, the following molecules can be selected for analysis:

1. Gas molecule: under the partial oxidation of GDY, gas molecules such as NO, O₂, NH₃, etc. can selectively react with it. For example, NH₃ can form hydrogen bonds with π electrons on the surface of GDY, and cause charge transfer and electronic structure change. Therefore, GDY can serve as a sensor for these gas molecules (1).
2. Organic molecules: alkenes and aromatic ring compounds have different electronic structures. When the surface of GDY interacts with the target molecule, π -electrons interact between them, resulting in changes in charge density. GDY can achieve selective response based on their charge density (2-4).
3. Metal ions: after various metal ions are loaded on GDY, its electronic structure changes, which can be used to detect and identify the existence and concentration of some metal ions (5).

The relevant literature has been supplemented in the revised manuscript. In the future, we will try to explore the chemoselectivity sensing properties of GDY according to the suggestions of reviewer.

References

- (46) Wu, Y., Chen, X., Weng, K., Arramel, J., Ong, W. J., Zhang, P., Zhao, X., Li, N., Highly Sensitive and Selective Gas Sensor Using Heteroatom Doping Graphdiyne: A DFT Study. *Adv. Electron. Mater.* **2021**, *7*, 2001244.
- (47) Zhang, X., Zhu, M., Chen, P., Li, Y., Liu, H., Li, Y., Liu, M. Pristine graphdiyne-hybridized photocatalysts using graphene oxide as a dual-functional coupling reagent. *Phys. Chem. Chem. Phys.* **2015**, *17*, 1217-1225.
- (48) Hussain, T., Sajjad, M., Singh, D., Bae, H., Lee, H., Larsson, J., Ahuja, R., Karton, A., Sensing of Volatile Organic Compounds on Two-Dimensional Nitrogenated Holey Graphene, Graphdiyne, and Their Heterostructure, *Carbon* **2020**,

163, 213-223.

(49) Yao, H. Y., Zhao, Y., Yang, N. L., Hao, W., Zhao, H., Li, S. Z., Zhu, J., Shen, L., Fang, W. H., Molecule Functionalization to Facilitate Electrocatalytic Oxygen Reduction on Graphdiyne, *J. Energy Chem.* **2022**, *65*, 141-148.

(50) Li, Y., Huang, H., Cui, R. L., Wang, D. M., Yin, Z., Wang, D., Zheng, L. R., Zhang, J., Zhao, Y. D., Yuan, H., Dong, J. Q., Guo, X. H., Sun, B. Y., Electrochemical Sensor based on Graphdiyne is Effectively Used to Determine Cd²⁺ and Pb²⁺ in Water. *Sens. Actuators B Chem.* **2021**, *332*, 129519.

Reviewer #2

They answered my questions sincerely and made reasonable inferences, although the mechanism of the Raman enhancement effect is not yet 100% certain. Therefore, I think it would be acceptable for this paper to be published in Nature Communications.

Reply: We are very grateful for the positive comment from the reviewer on our revision.

Reviewer #3

I am still concerned about the enhancement mechanism for BPA and 2,4-DCP. Their additional DFT calculation in this round of revision is too ambiguous and only shows the charge-density distributions between the analytes and GDY, which provides little evidence of ICT. Based on their DFT calculations, the authors should provide ICT pathways similar to those shown in Figure 4b and clarify their relations with the photon energy of the excited light. I cannot recommend the publication of this manuscript if the authors cannot provide a convincing explanation of the enhancement mechanism for BPA and 2,4-DCP.

Reply: We appreciate the reviewer's insightful and constructive comments and advice, and we have carefully addressed these concerns and made a proper revision of the manuscript. These comments and suggestions have not only enabled us to provide a highly improved manuscript but also inspired us to conduct more in-depth studies on SERS in future works.

Based on the suggestions of the reviewer, we further calculated the HOMO and LUMO energy levels of 2,4-DCP and BPA, and provided possible ICT pathways between them and GDY. For 2,4-DCP, the HOMO and LUMO energy levels are -5.73 eV and -1.90 eV, respectively. From the point of view of energy matching, it can be expected that contributions from three types of thermodynamically feasible ICT resonance may be related to the overall CM enhancement in the GDY-DCP system at the excitation of 532 nm (2.33 eV), including the exciton resonance of GDY (μ_{ex} , 1.71 eV), the photon induced charge transfer resonance (μ_{ICT} : 2.12 eV) together with the ground-state charge transfer resonance (μ_{GSCT} , 0.41 eV) from matched energy level between GDY and 2,4-DCP molecules.

For BPA, the HOMO and LUMO energy levels are -5.19 eV and -1.35 eV, respectively. Similar to 2,4-DCP, three types of thermodynamically feasible ICT resonance were found in GDY-BPA system, including the exciton resonance of GDY (μ_{ex} , 1.71 eV), the photon induced charge transfer resonance (μ_{ICT} : 1.58 eV) together with the ground-state charge transfer resonance (μ_{GSCT} , 0.13 eV).

We have added these discussions to the revised manuscript, and we believe that these contents will further enhance readers' understanding of the SERS properties of GDY. We sincerely thank the reviewer for the constructive suggestions.

Page 17 (yellow section): We also explored possible chemical enhancement mechanisms in GDY-2,4-DCP and GDY-BPA systems via density functional theory calculations. The interaction electrons distributed on the surface of GDY/2,4-DCP and GDY/BPA are shown in Figure S29, which indicates the strong chemical interactions between GDY substrate and probe molecules. For 2,4-DCP, the calculated HOMO and LUMO energy levels are -5.73 eV and -1.90 eV respectively. At the excitation of 532 nm (2.33 eV), it can be expected that contributions from three types of thermodynamically feasible ICT resonance may be related to the overall CM enhancement in the GDY-2,4DCP system, including the exciton resonance of GDY (μ_{ex} , 1.71 eV), the photon induced charge transfer resonance (μ_{ICT} : 2.12 eV), and the ground-state charge transfer resonance (μ_{GSCT} , 0.41 eV) from matched energy level between GDY and 2,4-DCP molecules (Figure S30). For BPA, the HOMO and

LUMO energy levels are -5.19 eV and -1.35 eV, respectively (Figure S31). Similar to 2,4-DCP, three types of thermodynamically feasible ICT resonance were also found in GDY-BPA system, including the exciton resonance of GDY (μ_{ex} , 1.71 eV), the photon induced charge transfer resonance (μ_{ICT} : 1.58 eV) together with the ground-state charge transfer resonance (μ_{GSCT} , 0.13 eV). The calculated results clearly reveal the ICT effect play a crucial role in SERS mechanism.

Figure S30. Band energy alignment diagram of the charge-transfer pathways in GDY-2,4-DCP system.

Figure S31. Band energy alignment diagram of the charge-transfer pathways in GDY-BPA system.

Reviewer #5

This revised paper can be published in Nature Communications without further revisions.

Reply: We are very grateful for the positive comment from the reviewer on our revision.

REVIEWER COMMENTS

Reviewer #1 (Remarks to the Author):

Authors addressed my questions and now manuscript can be accepted. I appreciate efforts of authors.

Reviewer #3 (Remarks to the Author):

I want to commend the authors' efforts in this round of revision and would like to recommend the publication of this manuscript if the authors can address my following comments on their new data shown in this round.

1. Based on the ICT pathways provided by the authors, the required photon energy for ICT between GDY HHMS and BPA is 1.58 eV, which corresponds to the wavelength of 785 nm. If their calculation is correct, a higher Raman signal enhancement can be expected with the excitation wavelength of 785 nm. Can the authors experimentally show Raman spectra and compare their enhancement factors at the excitation wavelengths of 532 nm and 785 nm?
2. The measured enhancement factors for PBA and 2,4-DCP should be calculated and given in the manuscript.

Reviewer #3

I want to commend the authors' efforts in this round of revision and would like to recommend the publication of this manuscript if the authors can address my following comments on their new data shown in this round.

Reply: We are very grateful for the reviewer's recognition of our efforts, and we also appreciate the the reviewer's constructive suggestions that have further improved the quality of our manuscript. Below, we will respond to the reviewer's questions one by one.

1. Based on the ICT pathways provided by the authors, the required photon energy for ICT between GDY HHMS and BPA is 1.58 eV, which corresponds to the wavelength of 785 nm. If their calculation is correct, a higher Raman signal enhancement can be expected with the excitation wavelength of 785 nm. Can the authors experimentally show Raman spectra and compare their enhancement factors at the excitation wavelengths of 532 nm and 785 nm?

Reply: This is a very good suggestion, and we have conducted a series of experiments to test this hypothesis. Interestingly, under the same experimental conditions, when we switched the excitation light to 785 nm, there was no significant change in the signal intensity of BPA compared to 532 nm. Figure S32 shows the Raman spectra of BPA at a concentration of 1×10^{-8} M under 532 nm and 785 nm excitation, respectively. It can be seen that the Raman peak intensity under 532 nm excitation is stronger than that under 785 excitation, but there is no difference of magnitude, which indicates that there is no significant difference in the enhancement effect generated by the two types of excitation. To further clarify this point, we further explored the Raman signal enhancement effects generated by two types of excitation at 1×10^{-11} M. The experimental results show that under the same laser power (0.7 mW) and integration time (40 s), the Raman enhancement effects generated by the two excitation are almost the same (Figure S33). The calculated Raman EFs are 2.4×10^6 (532 nm) and 1.9×10^6 (785 nm) at 1×10^{-10} M, respectively (the calculation process has been added to the Supporting Information).

Why is it inconsistent with our hypothesis? We carefully analyzed the possible

reasons. Although the energy of the 785 nm (1.58 eV) is highly consistent with that of the required photon energy for ICT between GDY HHMS and BPA, the 1.58 eV energy of 785 nm laser cannot initiate the excitation resonance of GDY (μ_{ex} , 1.71 eV). That is to say, compared to 532 nm excitation, there are only two forms of thermodynamically feasible resonance under 785 nm excitation. Therefore, overall, there is not much difference in the two types of excitation. The relevant discussion has been added to the revised manuscript (page 19).

Please See Manuscript (page 19, marked in yellow):

Interestingly, under the same experimental conditions, when we switched the excitation light to 785 nm (1.58 eV), there is no order of magnitude change in the signal intensity of BPA compared to 532 nm excitation. Figure S32 shows the Raman spectra of BPA at a concentration of 1×10^{-8} M under 532 nm and 785 nm excitation, respectively. It can be seen that the Raman peak intensity under 532 nm excitation is stronger than that under 785 nm excitation, but there is no difference of magnitude, which indicates that there is no significant difference in the enhancement effect generated by the two types of excitation. To further clarify this point, we further explored the Raman signal enhancement effects generated by two types of excitation at 1×10^{-10} M. The experimental results show that under the same laser power (0.7 mW) and integration time (40 s), the Raman enhancement effects generated by the two excitation are almost the same (Figure S33). The calculated Raman EFs are 2.4×10^6 (532 nm) and 1.9×10^6 (785 nm) at 1×10^{-10} M, respectively (the calculation process see Supporting Information). The possible reasons for this similarity are that although the energy of the 785 nm is highly consistent with that of the required photon energy (1.58 eV) for ICT between GDY HHMS and BPA, the energy of 785 nm excitation cannot initiate the excitation resonance of GDY (μ_{ex} , 1.71 eV). That is to say, compared to 532 nm excitation, there are only two forms of thermodynamically feasible resonance under 785 nm excitation (Figure S34). Therefore, overall, There is no significant difference in the enhancement effect between the two excitation.

Please See Supporting Information (page 2-3, marked in yellow):

1.2. Calculation of BPA Raman EF (532 nm excitation, 0.7 mW)

The calculation method of BPA EFs is consistent with the calculation method of R6G. Here we used the R6G peak at 846.7 cm^{-1} to estimate the EFs. The peak intensity at 846.7 cm^{-1} of BPA/GDY ($1.0 \times 10^{-10} \text{ M}$) is 4.75 counts with 40 s acquisition time, and that of bulk BPA is 215.1 counts with 0.5 s acquisition time. The density of bulk BPA is 1.19 g cm^{-3} , and the molar mass of BPA is $228.28 \text{ g mol}^{-1}$. Taking all the factors into account, the EF in the BPA/GDY measurements (532 nm) was derived as:

$$EF = \frac{I_{\text{SERS}}}{I_{\text{bulk}}} \times \frac{\rho h A_{\text{sub}}}{cVM} = 3.3 \times 10^{-4} \times \frac{1.19 \times 21 \times 10^{-4} \times 0.8}{1 \times 10^{-10} \times 10 \times 10^{-6} \times 228.28} = 2.4 \times 10^6$$

1.3. Calculation of BPA Raman EF (785 nm excitation, 0.7 mW)

The calculation method of BPA EFs is consistent with the calculation method of R6G. Here we used the BPA peak at 846.7 cm^{-1} to estimate the EFs. The peak intensity at 846.7 cm^{-1} of BPA/GDY ($1.0 \times 10^{-10} \text{ M}$) is 3.43 counts with 40 s acquisition time, and that of bulk BPA is 198.3 counts with 0.5 s acquisition time. The density of bulk BPA is 1.19 g cm^{-3} , and the molar mass of BPA is $228.28 \text{ g mol}^{-1}$. Taking all the factors into account, the EF in the BPA/GDY measurements (785 nm) was derived as:

$$EF = \frac{I_{\text{SERS}}}{I_{\text{bulk}}} \times \frac{\rho h A_{\text{sub}}}{cVM} = 2.2 \times 10^{-4} \times \frac{1.19 \times 21 \times 10^{-4} \times 0.8}{1 \times 10^{-10} \times 10 \times 10^{-6} \times 228.28} = 1.9 \times 10^6$$

Figure S32. SERS spectra of BPA under 532 nm and 785 nm excitation. BPA concentration: $1 \times 10^{-8} \text{ M}$, laser power: 0.7 mW, integration time: 3 s.

Figure S33. SERS spectra of BPA under 532 nm and 785 nm excitation. BPA concentration: 1×10^{-10} M, laser power: 0.7 mW, integration time: 40 s.

Figure S34. Band energy alignment diagram of the charge-transfer pathways in BPA/GDY HHMSs under different excitation.

2. The measured enhancement factors for BPA and 2,4-DCP should be calculated and given in the manuscript.

Reply: This is a very good suggestion. In the first question, we have provided the Raman EFs and calculation process of BPA under 532 nm and 785 nm excitation, respectively. Next, based on the suggestion of the reviewer, we calculated the Raman EFs of 2,4-DCP under 532 nm and 785 nm excitation. The calculation results indicate that under 532 nm excitation, the Raman EF obtained is 4.3×10^6 . But when 532 nm excitation was replaced with 785 nm excitation, no distinguishable 2,4-DCP signals was found, except for the scattering signals of GDY itself. This is because the energy

excited by 785 nm (1.58 eV) is too low to drive the exciton resonance of GDY (μ_{ex} , 1.71 eV) or initiate interface ICT resonance (μ_{ICT} : 2.12 eV). Therefore, under 785 nm excitation, only one weak ground-state charge transfer resonance (μ_{GSCT} , 0.41 eV) exists, which cannot produce a distinguishable Raman spectrum. We have added the Raman EF of 2,4-DCP (under 532 excitation) in the revised manuscript, and the relevant calculation process has been supplemented in the Supporting Information.

Please See Manuscript (page 20, marked in yellow): We also calculated the Raman EFs of 2,4-DCP under 532 nm and 785 nm excitation. The calculation results indicate that under 532 nm excitation, the Raman EF obtained is 4.3×10^6 (the calculation process see Supporting Information). But when 532 nm excitation was replaced with 785 nm excitation, no distinguishable 2,4-DCP signals was found, except for the fluorescent background (Figure S35). This is because the energy excited by 785 nm (1.58 eV) is too low to drive the exciton resonance of GDY (μ_{ex} , 1.71 eV) or initiate interface ICT resonance (μ_{ICT} : 2.12 eV). Therefore, under 785 nm excitation, only one weak μ_{GSCT} resonance (0.41 eV) exists, which cannot produce a distinguishable Raman spectrum.

Figure S35. SERS spectrum of 2,4-DCP under 785 nm excitation. 2,4-DCP concentration: 1×10^{-10} M, laser power: 0.7 mW, integration time: 40 s.

Please See Supporting Information (page 2-3, marked in yellow):

1.4. Calculation of 2,4-DCP Raman EF (532 nm excitation, 0.7 mW)

The calculation method of 2,4-DCP EFs is consistent with the calculation method of R6G. Here we used the 2,4-DCP peak at 865.2 cm⁻¹ to estimate the EFs. The peak intensity at 865.2 cm⁻¹ of 2,4-DCP/GDY (1.0×10⁻¹⁰ M) is 5.16 counts with 40 s acquisition time, and that of bulk 2,4-DCP is 205.2 counts with 0.5 s acquisition time. The density of bulk 2,4-DCP is 1.38 g cm⁻³, and the molar mass of BPA is 163 g mol⁻¹. Taking all the factors into account, the EF in the 2,4-DCP/GDY measurements (532 nm) was derived as:

$$EF = \frac{I_{\text{SERS}}}{I_{\text{bulk}}} \times \frac{\rho h A_{\text{sub}}}{cVM} = 3.02 \times 10^{-4} \times \frac{1.38 \times 21 \times 10^{-4} \times 0.8}{1 \times 10^{-10} \times 10 \times 10^{-6} \times 163} = 4.3 \times 10^6$$

REVIEWERS' COMMENTS

Reviewer #3 (Remarks to the Author):

The authors have addressed my comments. I recommend its publication.